physiology/evolution/ecology

acclimation response, critical thermal maximum, standard metabolic rate, cardiac metabolic rate, cardiac remodeling, thermal tolerance

**Author for correspondence:**
Melissa K. Drown
e-mail: mxd1288@miami.edu

# Interindividual plasticity in metabolic and thermal tolerance traits from populations subjected to recent anthropogenic heating

Melissa K. Drown, Amanda N. DeLiberto,

Moritz A. Ehrlich, Douglas L. Crawford and

Marjorie F. Oleksiak

Rosenstiel School of Marine and Atmospheric Science, University of Miami, Miami, FL, USA

MKD, 0000-0002-7350-984X; AND, 0000-0001-7055-8091;
MAE, 0000-0002-0774-2037

To better understand temperature's role in the interaction between local evolutionary adaptation and physiological plasticity, we investigated acclimation effects on metabolic performance and thermal tolerance among natural *Fundulus heteroclitus* (small estuarine fish) populations from different thermal environments. *Fundulus heteroclitus* populations experience large daily and seasonal temperature variations, as well as local mean temperature differences across their large geographical cline. In this study, we use three populations: one locally heated (32°C) by thermal effluence (TE) from the Oyster Creek Nuclear Generating Station, NJ, and two nearby reference populations that do not experience local heating (28°C). After acclimation to 12 or 28°C, we quantified whole-animal metabolic (WAM) rate, critical thermal maximum (CT$_{max}$) and substrate-specific cardiac metabolic rate (CaM, substrates: glucose, fatty acids, lactate plus ketones plus ethanol, and endogenous (i.e. no added substrates)) in approximately 160 individuals from these three populations. Populations showed few significant differences due to large interindividual variation within populations. In general, for WAM and CT$_{max}$, the interindividual variation in acclimation response (log$_2$ ratio 28/12°C) was a function of performance at 12°C and order of acclimation (12–28°C versus 28–12°C). CT$_{max}$ and WAM were greater at 28°C than 12°C, although WAM had a small change (2.32-fold) compared with the expectation for a 16°C increase in temperature (expect 3- to 4.4-fold). By contrast, for CaM, the rates when acclimatized and assayed at 12 or 28°C

were nearly identical. The small differences in CaM between 12 and 28°C temperature were partially explained by cardiac remodeling where individuals acclimatized to 12°C had larger hearts than individuals acclimatized to 28°C. Correlation among physiological traits was dependent on acclimation temperature. For example, WAM was negatively correlated with $CT_{max}$ at 12°C but positively correlated at 28°C. Additionally, glucose substrate supported higher CaM than fatty acid, and fatty acid supported higher CaM than lactate, ketones and alcohol (LKA) or endogenous. However, these responses were highly variable with some individuals using much more FA than glucose. These findings suggest interindividual variation in physiological responses to temperature acclimation and indicate that additional research investigating interindividual may be relevant for global climate change responses in many species.

# 1. Introduction

The ability to respond to changing temperature associated with climate change can include acclimation and evolved responses. Along the Atlantic coast of North America, *Fundulus heteroclitus* populations experience temperatures that differ by greater than 14°C, this small estuarine fish has been a focus of metabolic and biochemical studies investigating temperature responses [1,2]. Yet even within a single estuary, a *F. heteroclitus* population experiences high variability in multiple abiotic factors [1–3]: temperatures can increase daily to greater than 30°C in upper estuaries or plunge to 12–15°C with incoming cold tides [4]; salinity can vary from nearly freshwater due to heavy rains to salinities greater than seawater (greater than 30 ppt) in desiccating ponds [3,5–7]; oxygen concentrations can vary from anoxic to supersaturated [7]. Seasonally, populations in the northern part of the range may additionally experience temperatures that vary by greater than 20°C from summer to winter with some populations probably experiencing freeze–thaw cycles during the winter months. *Fundulus heteroclitus*' ability to mount physiological responses to tolerate these variable conditions is well documented [1,6,8–13], and these variable conditions within and among *F. heteroclitus* populations are thought to drive resulting phenotypic differences via physiological plasticity and evolutionary adaptation [2].

Physiological plasticity or acclimation responses, defined as an active physiological response to environmental change that alters phenotype, has long been a subject of debate from an evolutionary perspective [14,15]. Specifically, acclimation can alter physiological processes to modulate the effect of environmental change [14]. Plasticity in general, and acclimation response specifically, can result in similar phenotypes despite differing underlying genotypes, this can cause genetic differences in performance to be 'hidden' from evolutionary forces [15]. This results in two contrasting views on how acclimation response affects evolutionary adaptation. First, it might hinder evolution by masking maladaptive genetic variation and allowing individuals to persist who do not have the optimum phenotype but do have an acclimation response. By contrast, physiological acclimation might enhance evolution by allowing individuals to persist in a novel environment long enough for evolution to act, thus making those populations well suited for adaptation [16–19]. This latter point may be especially important for populations adapting to highly variable or rapidly changing environments where short-term physiological responses are key in allowing individuals to survive long enough to reproduce [20,21]. While these views differ in how physiological acclimation and evolutionary adaptation interact, they are not necessarily mutually exclusive pathways for organismal survival. If physiological acclimation allows for higher average fitness across all experienced environments, then individuals with a greater acclimation response may be selected for in variable and rapidly changing environments. Alternatively, if acclimation responses have associated cost (e.g. lower fecundity or survival with greater response [22]), a highly variable environment may favour the maintenance of variation in acclimation responses [9,23]. Yet, for animals, there is little information on the interindividual variation in acclimation responses, how, or whether this variation among traits is related, and the potential evolutionary importance (although see [24] and plant literature, e.g. [25–27]. Here, we examine the variation in physiological performance and how this relates to acclimation within and between traits, concluding that large interindividual variation in physiological performance at low temperatures alters the magnitude of acclimation response.

To better understand the role of physiological acclimation and evolutionary adaptation, we investigated how temperature acclimation affects physiological performance among approximately 160 individuals from three wild *F. heteroclitus* populations that experience different local temperatures. Six physiological traits known to be temperature-sensitive—whole-animal metabolic rate (WAM), critical thermal maximum ($CT_{max}$) and cardiac metabolic rate (CaM, oxygen consumption of heart ventricles in

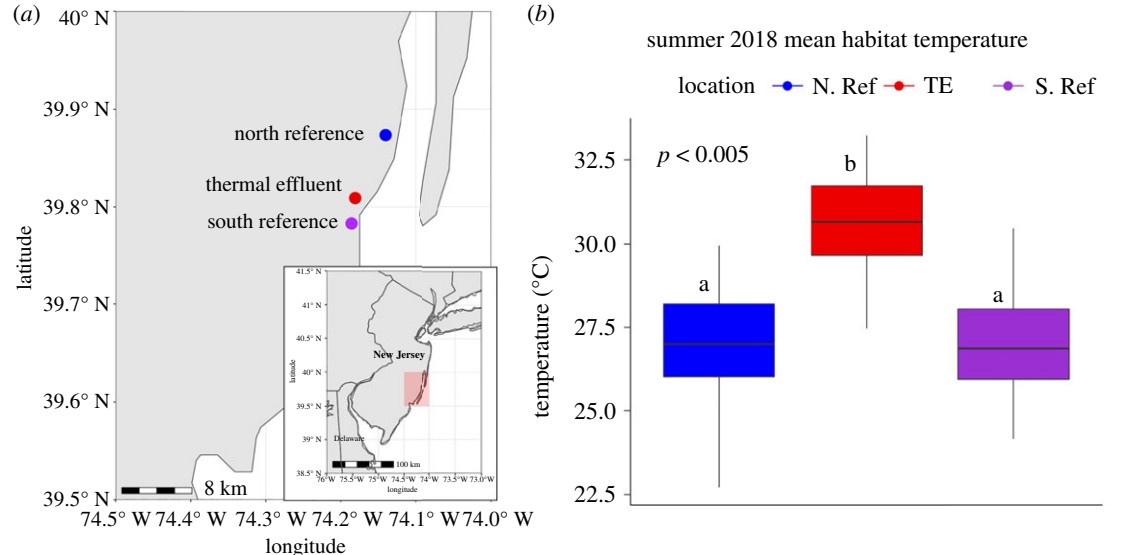

**Figure 1.** Oyster Creek triad. (*a*) Oyster Creek triad in New Jersey, USA, with north reference (blue, N. Ref), south reference (purple, S. Ref) and effluent site (red, TE). The Forked River connects the Oyster Creek Nuclear Generating Station (OCNGS) to Barnegat Bay, and water from the bay is used for nuclear reactor cooling. (*b*) Temperature data collected in summer 2018 using HOBO loggers. The TE site is significantly warmer (4°C) than both reference sites (ANOVA, $p < 0.01$). Mean summer temperatures: TE 32°C, N. Ref and S. Ref 28°C.

the presence of glycolytic and non-glycolytic aerobic substrates)—were measured in individuals from three populations [28]. For CaM, we measured heart tissue oxygen consumption in the presence of four aerobic substrates: glucose, fatty acids, lactate plus ketones plus ethanol, and endogenous (i.e. non-glycolytic metabolism with no added substrates) [29–31]. Populations include one experiencing local heating by thermal effluence (TE) from a nuclear power plant and two reference populations that do not experience local heating (10 km north and 3.5 km south of the TE population, figure 1*a*). Local anthropogenic heating is caused by the Oyster Creek Nuclear Generating Station (OCNGS), which has operated on the east coast of New Jersey, USA, since 1969 and is positioned along the Forked River inside Barnegat Bay. Using these geographically close Oyster Creek populations allows neutral divergence due to demography to be distinguished from potentially adaptive divergence due to local temperature variation, which could include variation in acclimation response [8,32–34]. Prior population genetic analyses suggest that the Oyster Creek TE population is locally adapted compared with reference populations in southern and northern New Jersey [32].

Here, we examine six physiological traits at two acclimation temperatures to determine (i) the variation within and among populations, (ii) how these traits are affected by acclimation, (iii) the correlations among traits, and (iv) the relationships between the magnitude of the acclimation response and the interindividual variation in physiological performance. The results show few differences among populations, high interindividual variation for each trait depending on the acclimation temperature, interindividual variation in acclimation responses among traits and significant temperature-dependent correlations among traits. These data suggest that the acclimation responses are largely defined by individual performance at 12°C, with individuals that have low $CT_{max}$ or WAM at 12°C having greater acclimation responses. Assuming a reasonable heritability for these six traits and acclimation responses, as suggested by other studies [8,22,35–39], the data presented here suggest that evolution favours the maintenance of interindividual variation in both physiological performance at a specific temperature and the magnitude of acclimation response to temperature change.

# 2. Methods

## 2.1. Animal care and use

All fish were caught in live traps in New Jersey, USA, in September 2018 at three sites: north reference (N. Ref; 39°52′28.0″ N, 74°08′19.0″ W), thermal effluent (TE; 39°48′33.0″ N, 74°10′51.0″ W) and south

reference (S. Ref; 39°47′04.0″ N, 74°11′07.0″ W) and transported live to the University of Miami where they were housed according to the University of Miami Institutional Animal Care and Use Committee guidelines (Animal Use Protocol No: 16-127-adm04). From July to September 2018, HOBO data loggers were placed at all three sites and used to collect temperature data at a rate of one measure every 5 min. Fish were common gardened at 20°C and 15 ppt for more than six weeks on a summer light cycle (14 h daylight, 10 h dark), then overwintered at 10°C and 15 ppt (5 h daylight 19 h dark) for four weeks. Here, common gardening refers to acclimation to common temperature, salinity and light cycle to remove the reversible effects of acclimation to local environmental conditions, which may be present in individuals collected from different populations. All individuals were uniquely tagged using subdermal visual implant elastomer (VIE) injections. Individuals from each site were then either acclimatized to 12 or 28°C and 15 ppt acclimation conditions on a summer light cycle for four weeks. After the first acclimation period (12 or 28°C), whole-animal metabolism was measured followed by a one-week recovery period before measuring $CT_{max}$. Following whole-animal metabolism and $CT_{max}$ measurements, individuals were acclimatized to the alternative temperature for at least four weeks and both physiological measures repeated. After a recovery period of at least two weeks post-critical thermal maximum determinations, CaMs were measured. Fish were fed pelleted food to saturation once daily (EP1 diet, Marubeni Nisshin Feed Co., Chuo-ku, Tokyo) and fasted for 24 h prior to any phenotypic measurement. *Fundulus heteroclitus* were collected on public lands and do not require a permit for non-profit use.

## 2.2. Whole-animal metabolism

WAM rates were measured as oxygen consumption rate for each individual at both 12°C when acclimatized to 12°C and at 28°C when acclimatized to 28°C. Oxygen consumption was quantified using a high-throughput intermittent flow respirometer (HIFR) [40].

Individuals were measured overnight where they were left undisturbed for at least 14 h. At least 25 measurement periods (6 min at 28°C and 12 min at 12°C (measurement periods were longer at 12°C due to the longer time needed to achieve approx. 10% decrease in oxygen)) were recorded for each individual overnight, and the slope of oxygen levels over time was extracted using a linear model for each replicate measurement period. Of those, at least 20 replicates were used for analysis, with a few values excluded based on low $R^2$ value (minimum $R^2 = 0.9$). Metabolic rate ($MO_2$, mg $O_2$ $h^{-1}$) was calculated with $MO_2 = KV$, where $K$ is the slope (µmol $O_2$ $min^{-1} l^{-1}$), and $V$ is the volume of the respirometer minus volume of the organism (litres), and units were converted to mg $O_2$ $h^{-1}$ [41]. To capture a minimum or resting metabolic rate (standard metabolic rate, SMR), a single value was defined by the 10th percentile values from the cumulative frequency distribution (CFD) of all replicate metabolic rates from each individual (minimum 20 replicate rates of oxygen consumption per time). This 10th percentile value captures the time period when the fish were most at rest during measurement and excludes the lowest tail of the data distribution by selecting a value for SMR that lies on the CFD curve rather than averaging the lowest 10% of data points, which may be sensitive to outliers [42]. Further description of data collection and analysis from raw PreSens datafiles through metabolic rate calculation including correcting for background respiration can be found in the published methods manuscript [40].

## 2.3. Critical thermal maximum

Critical thermal maximum ($CT_{max}$) was measured one week after metabolic rate determinations. $CT_{max}$ was measured using a 10-gallon aquarium filled with fully oxygenated seawater (15 ppt) at the acclimation temperature and heated at 0.3°C $min^{-1}$ using a submersible heating coil. The rate of temperature change is consistent with other published studies and prevents lag between body and water temperature during $CT_{max}$ measurement [43]. The temperature at which fish had no coordinated movement for at least 5 consecutive seconds was recorded as the $CT_{max}$. Individuals were placed in 12°C recovery beakers for 1 h following $CT_{max}$ measurements to mitigate negative effects of heat stress before being returned to the appropriate acclimation condition.

## 2.4. Cardiac metabolism

Substrate-specific CaM was measured in a custom chamber system [28]. To obtain CaM measurements, fish were sacrificed via cervical dislocation, and hearts were immediately removed and placed in Ringer's

media (1.5 mM $CaCl_2$, 10 mM Tris–HCl pH 7.5, 150 mM NaCl, 5 mM KCl, 1.5 mM $MgSO_4$) with 5 mM glucose and $10\,U\,ml^{-1}$ heparin to expel blood. Ventricles were splayed open to expose the inner ventricular surface and limit muscular contractions. To control temperature, a custom external Plexiglas water bath with four 1 ml micro-respiration chambers (UNISENSE, Aarhus, Denmark) was used [28]. Each chamber contained a micro stir bar and nylon mesh screen. A fluorometric oxygen sensor spot (PreSens Precision Sensing, Regensburg, Germany) was adhered to the internal side of the chamber lid. A fibre-optic cable was affixed to each chamber lid for contactless oxygen measurement through the sensor spot, and all cables were connected to a 10-channel oxygen meter (PreSens Precision Sensing, Regensburg, Germany). Oxygen consumption over time was used to calculate CaM. PreSens Measurement Studio 2 software was used to collect oxygen data.

Four separate substrates were used for CaM: (i) GLU—5 mM glucose, (ii) FA—fatty acids (1 mM Palmitic acid conjugated to fatty-acid-free bovine serum albumin), (iii) LKA—lactate ketones and alcohol (5 mM lactate, 5 mM hydroxybutyrate, 5 mM ethyl acetoacetate, 0.1% ethanol), and (iv) END—substrate-free Ringer's media (non-glycolytic endogenous metabolism). These substrate concentrations are commonly used for assaying substrate metabolism in teleost ventricles [34,44–46]. After CaM GLU measurements were taken, glycolytic enzyme inhibitors (20 mM 2-deoxyglucose and 10 mM iodoacetate) were added so that glucose-independent measures using the other three substrates could be taken [34,44–46]. To quantify CaM, the oxygen consumption of the heart was recorded for 6 min in the presence of each of four substrates separately and in order (GLU, FA, LKA, END). Oxygen consumption rate during the last 3 of the 6 min was used to calculate CaM in pmol $O_2\,s^{-1}$. During each day of measurement, a minimum of three blank runs, during which only media were in the chamber, were recorded to determine any leak or background oxygen consumption within the chambers. Each CaM was corrected for background leak and oxygen consumption by subtracting the mean oxygen consumption rate of these three blank runs from the substrate-specific metabolic rate measured in that chamber [28]. Individual chambers were used for only one substrate with hearts rotated among chambers to measure substrate-specific metabolic rate.

## 2.5. Statistical analysis

All analyses were performed in R (v. 3.6) and verified in SAS-JMP (SAS Institute, Cary, NC, USA). All data are available at the Dryad Digital Repository (https://doi.org/10.5061/dryad.0gb5mkm0w) and all scripts used in analysis can be found on github: https://github.com/mxd1288/physiological_plasticity_funhe.

Acclimation response is defined as the $\log_2$ ratio of 28/12°C measures for an individual (i.e. a response equal to 0 indicates that measures at 12 and 28°C are the same). $\log_2$ transformed ratios were used to achieve a normal distribution. To compare variance across traits measured on different scales, the within-group coefficient of variation (CV) was calculated as $100\% \times$ standard deviation divided by mean. To compare individuals that differed in body mass for metabolic rates, the regression of trait value versus body mass was used to calculate body mass residuals. Additionally, to compare means and variance among traits without variance due to body mass, the mean trait value corrected for body mass was calculated as $X_{corrected} = X_{mean} + Y$, where $X_{mean}$ is the predicted trait value for an average size individual (10.2 g) from the trait versus body mass regression, and $Y$ are body mass residuals for each individual. Note that exact sample sizes for each substrate within each temperature for cardiac metabolism vary slightly as some individuals were removed from the analysis due to technical error (electronic supplementary material, table S1). Variation in sample size among phenotypes was primarily due to $CT_{max}$ survival (91.7% survived), consistent with previously reported $CT_{max}$ survival rates [24,30], and removal from the study due to loss of health (severe loss of body mass or changes in behaviours). In total, 163 individuals were measured for at least one physiological trait.

Within each acclimation temperature, there are two groups depending on whether they were acclimatized to 12°C first then 28°C or vice versa: group 1 that was acclimatized to and assayed at 12°C first before being acclimatized to and assayed at 28°C and group 2 that was acclimatized and assayed at 28°C first before being acclimatized and assayed at 12°C. Groups contained approximately equal numbers of males and females from all three populations with an even size distribution. The variable 'acclimation order' captures variance between groups within an acclimation temperature. Linear mixed models including body mass, acclimation temperature, acclimation order (group 1 and group 2), sex and population with all possible second-order interaction terms were used to determine the model of best fit for each physiological trait including plasticity in WAM and $CT_{max}$ [47] (electronic supplementary material, table S1). For CaMs, substrate was also included as a covariate, but acclimation

order was excluded because all individuals assayed at a single acclimation temperature for CaM were from the same acclimation order group. The details of these models, sample sizes, CV and significance of all terms are available in electronic supplementary material, table S1. To examine relationships among traits, Pearson's partial correlation coefficients were calculated. Partial correlation coefficients are the correlations between two physiological traits controlling for the covariance among the other traits [48]. For all traits, these partial correlations are not due to body mass because the residuals from mass-trait regressions were used to calculate partial correlation coefficients.

# 3. Results

## 3.1. Significant environmental temperature differences due to local anthropogenic heating

Data collected using HOBO data loggers deployed July–September 2018 show that the mean summer water temperature at the TE site was significantly higher (32°C) than sites north (N. Ref) and south (S. Ref) of the effluent (28°C, figure 1b). Variance in the mean summer temperatures during these four months did not significantly differ among sites ($p > 0.05$). During the winter of 2018 (October 2018–May 2019), HOBO data loggers remained at all three sites; however, due to loss of two loggers, only data from the S. Ref were recovered. The S. Ref experienced a mean winter temperature (November–March) of 3.6°C with consistent dips below freezing. This indicates that this site could experience freeze–thaw periods during the winter with an annual range in temperature exceeding 20°C.

## 3.2. Whole-animal metabolic rate

The $\log_{10}-\log_{10}$ relationship between WAM rates and body mass was significant ($p < 0.001$) and explained 8–11% of variation among individuals for 28 and 12°C acclimation temperatures (table 1). Thus, all analyses used the residuals of $\log_{10}$-regression to correct for body mass. WAM, using $\log_{10}$ mass residuals, among individuals acclimatized and assayed at 12 and 28°C had no significant differences among populations. However, WAM was significantly higher at 28°C than 12°C ($p < 0.001$; electronic supplementary material, figure S1) and acclimation temperature explained 59.8% of variation among individuals. Acclimation to 28°C relative to 12°C increased WAM by 2.32-fold (95% CI 2.1–2.8-fold), resulting in a $Q_{10}$ (fold change for every 10°C change) of 1.68 after correcting for body mass.

Acclimation order had a significant interaction with acclimation temperature ($p < 0.001$; electronic supplementary material, table S1). For WAM, this interaction reflects the effect of previous acclimation on the 12°C WAM determinations (electronic supplementary material, figure S1). For 12°C acclimation individuals, acclimation order had a significant effect with higher metabolic rates for individuals in group 1 (12°C first, then 28°C) than group 2 (28°C first, then 12°C; $p < 0.001$). Acclimation order had a small and insignificant difference between groups when individuals were acclimatized to 28°C (electronic supplementary material, figure S1).

While nearly all individuals had an increased WAM at 28°C compared with 12°C acclimation conditions, there was variation among individuals in the degree of WAM acclimation response (data at both acclimation temperatures for WAM, $N = 58$). That is, WAM acclimation response (here as the $\log_2$ ratio 28/12°C) varied among individuals with 28% of individuals showing little acclimation compensation (greater than threefold increase between temperatures) and three individuals having nearly perfect acclimation compensation with nearly the same WAM at 12 and 28°C ($\log_2$ ratio $\sim 0$, figure 2c). This range of acclimation responses for WAM arises because individuals with low 12°C metabolic rates had high 28°C metabolic rates and individuals with high 12°C metabolic rates had low 28°C metabolic rates (figure 2a,b). This is most clearly seen when the acclimation response is divided into three groups: (i) low—individuals in the bottom 10% confidence interval (CI), (ii) high—individuals in the top 10% CI, and (iii) most individuals with average acclimation response, between 10 and 90% CI. Individuals in the bottom 10% CI of acclimation response ($\log_2$ WAM 28°C/12°C) had high 12°C WAM but low 28°C WAM, and thus low acclimation response. Individuals in the top 10% CI had low 12°C WAM and high 28°C WAM and thus large acclimation response. For most individuals (between 10 and 90% CI), the WAM at 12 and 28°C were significantly positively correlated ($R^2 = 0.30$, $p < 0.0001$), thus, the magnitude of the acclimation response is a function of the WAM at each temperature. Acclimation order also had a significant effect on plasticity in WAM and explained 8.12% of variation among individuals leaving a substantial proportion of variation among

**Table 1.** Trait-specific means. Mass independent WAM, whole-animal metabolism (mg $O_2$ h$^{-1}$), CaM (pmol $O_2$ s$^{-1}$), cardiac metabolism with specific substrates (FA, fatty acids; GLU, glucose; LKA, lactate, ketones and ethanol; and END, no substrates endogenous) and CT$_{max}$ (°C), critical thermal maximum. For each trait at each acclimation temperature: $N$, mean, standard deviation of mean (s.d.), mean corrected for variation in mass,[a] standard deviation of mass-corrected mean (s.d.), coefficient of variation (CV = $100 \times$ s.d./mean) and $R^2$ for relationship with mass.

| trait | acd. temp | $N$ | mean | s.d. of mean | mean (corrected for mass) | s.d. of mass-corrected mean[a] | CV % | mass $R^2$ | 28/12°C | $Q_{10}$ |
|---|---|---|---|---|---|---|---|---|---|---|
| WAM | 12 | 85 | 1.86 | 0.69 | 1.82 | 0.66 | 36.31 | 0.1138 | 2.32 | 1.68 |
| WAM | 28 | 112 | 4.27 | 1.31 | 4.23 | 1.25 | 29.51 | 0.0775 | | |
| LogWAM | 12 | 85 | −0.21 | 0.16 | −0.93 | 0.66 | −71.18 | 0.1124 | −1.59 | n.a. |
| LogWAM | 28 | 112 | 0.16 | 0.13 | 1.48 | 1.25 | 84.32 | 0.0799 | | |
| CaM_FA | 12 | 50 | 35.35 | 21.87 | 34.45 | 18.73 | 54.37 | 0.2664 | 0.97 | 0.95 |
| CaM_FA | 28 | 49 | 32.82 | 9.81 | 33.31 | 7.36 | 22.09 | 0.4414 | | |
| CaM_GLU | 12 | 58 | 42.77 | 17.5 | 42.52 | 15.87 | 37.32 | 0.1810 | 1.11 | 1.09 |
| CaM_GLU | 28 | 51 | 48.84 | 14.41 | 47.11 | 12.24 | 25.97 | 0.3028 | | |
| CaM_LKA | 12 | 58 | 29.64 | 8.59 | 29.47 | 7.38 | 25.05 | 0.2907 | 0.98 | 0.98 |
| CaM_LKA | 28 | 50 | 28.85 | 10.1 | 28.82 | 8.14 | 28.25 | 0.4241 | | |
| CaM_END | 12 | 55 | 28.2 | 11.92 | 27.73 | 10.52 | 37.93 | 0.2261 | 0.65 | 0.76 |
| CaM_END | 28 | 49 | 18.12 | 8.55 | 17.98 | 6.9 | 38.38 | 0.3693 | | |
| CT$_{max}$ | 12 | 114 | 36.28 | 0.81 | 26.3 | 0.79 | 3.0 | 0.0555 | 1.23 | 1.10 |
| CT$_{max}$ | 28 | 97 | 42.47 | 0.27 | 32.48 | 0.36 | 1.1 | 0.0020 | | |

[a]Mass-corrected value = $X_{mean} + Y$, where $X_{mean}$ is the predicted trait value for an averaged size individual (10.2 g) from the trait versus body mass regression and $Y$ are body mass residuals for each individual.

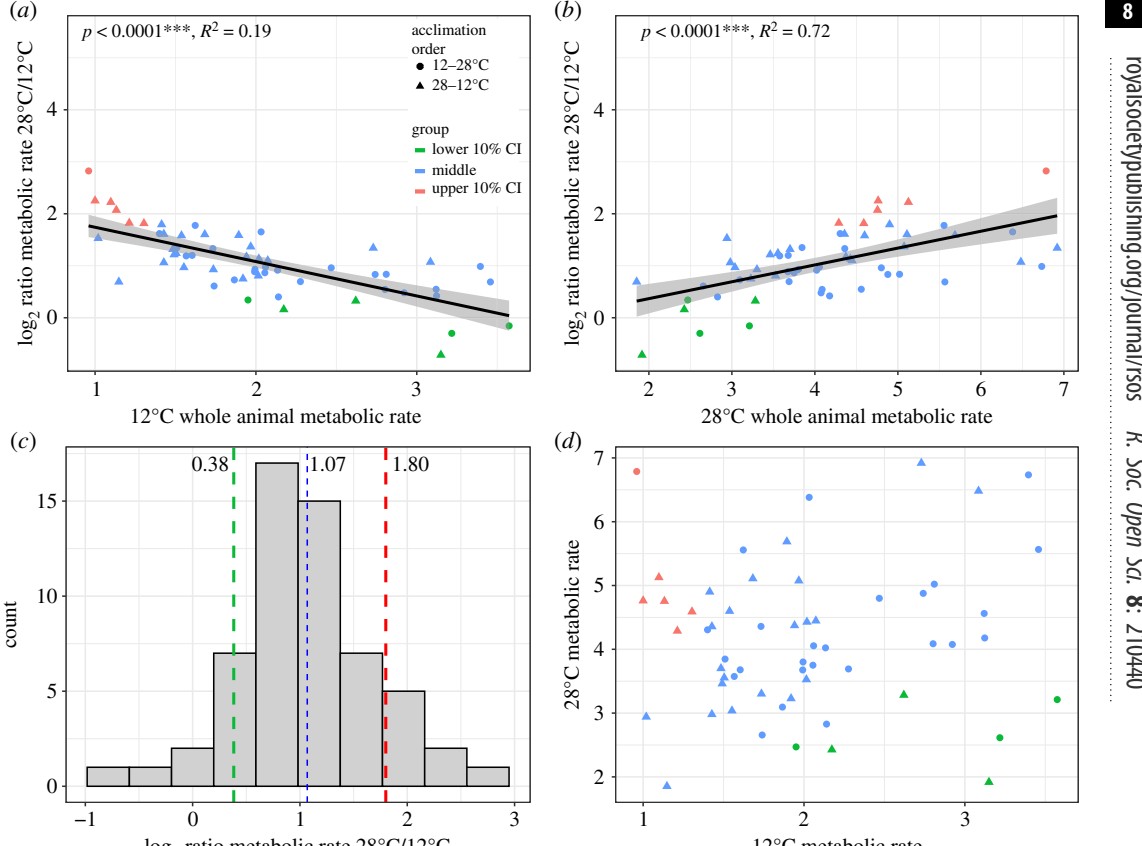

**Figure 2.** Plasticity in WAM rate. (*a*) Plasticity (log$_2$ difference between 28 and 12°C) in WAM rate was significantly correlated with both 12°C and (*b*) 28°C metabolic rate, with a negative correlation between 12°C metabolic rate and plasticity and a positive correlation between 28°C metabolic rate and plasticity, $N = 58$. (*c*) The distribution of the log$_2$ ratio 28/12°C whole metabolic rates with dashed line indicating the 90% confident interval. (*d*) Plot of 12°C WAM rates versus 28°C metabolic rates. Individuals (red) with both a low 12°C metabolic rate and high 28°C metabolic rate had the greatest plasticity while individuals with a higher 12°C metabolic rate (green) had lower plasticity. *p*-values from linear regressions; shaded region shows 95% confidence interval. Acclimation order had a significant effect on plasticity in WAM rate (shape in *a*, *b* and *d*: circles = 12–28°C, triangles = 28–12°C, $p < 0.05$).

individuals unexplained by the variables measured here ($p < 0.05$). However, the observed groupings of individuals based on variation in plasticity (three groups with high, intermediate or low acclimation response) do not appear to be driven by acclimation order. That is, not all individuals with high, low or intermediate plasticity were from one acclimation order group. Surprisingly, there was no overall correlation between 12 and 28°C metabolic rates, which one might expect since the metabolic activities at both acclimation temperatures were significantly related to plasticity (electronic supplementary material, figure S2).

## 3.3. Critical thermal maximum

Similar to WAM rate, CT$_{max}$ was significantly higher at 28°C than 12°C for all individuals ($p < 0.001$; electronic supplementary material, figure S3), and acclimation order (group) had a significant effect (data at both acclimation temperatures for CT$_{max}$, $N = 88$). Among individuals, acclimation temperature explained 95% of the CT$_{max}$ variation. The interaction between acclimation order and acclimation temperature ($p < 0.001$) was significant for 12°C acclimatized individuals (approx. 1°C difference) but not for 28°C acclimatized individuals (Tukey *post hoc* test). In contrast with the acclimation order effect for WAM rate, individuals in group 2 (acclimatized to 28°C first, then 12°C) had a greater CT$_{max}$ when measured at 12°C than those in group 1 (acclimatized to 12°C first, then to 28°C, $p < 0.001$). The CT$_{max}$ interindividual variation was also greater at 12°C than 28°C (CV = 2.22% at 12°C and 0.62% at 28°C) with individuals acclimatized to 12°C having a CT$_{max}$ range of

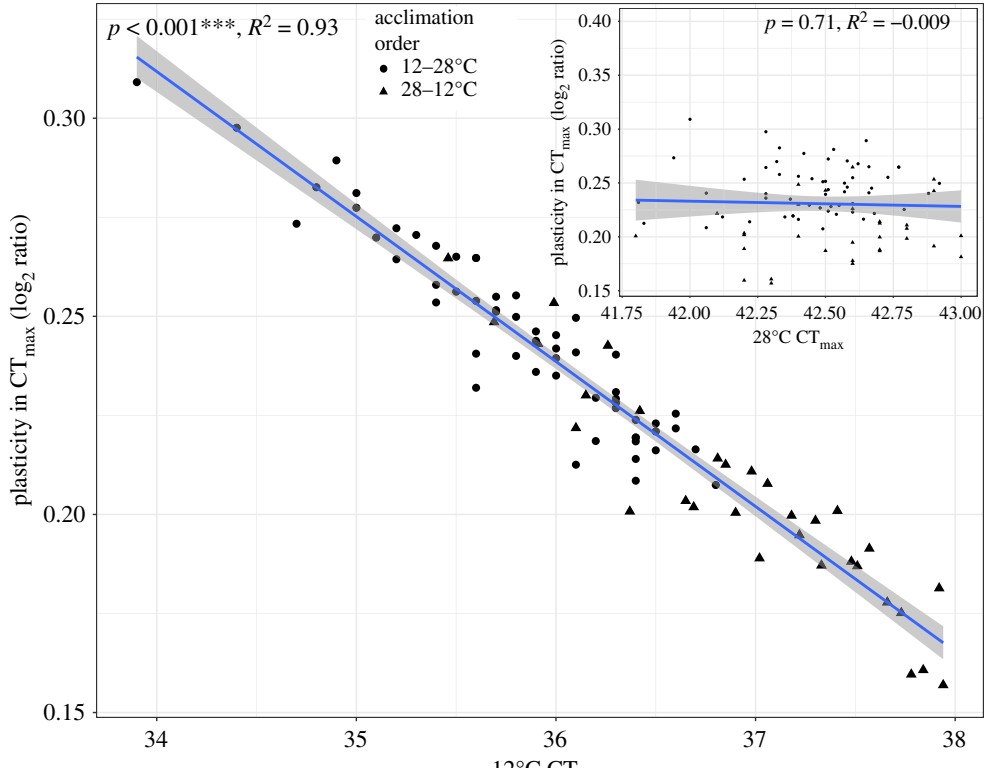

**Figure 3.** Plasticity of critical thermal maximum. Individuals with a greater ratio (higher plasticity) between 28 and 12°C $CT_{max}$ had a lower 12°C $CT_{max}$ while individuals with a smaller ratio (lower plasticity) had a higher 12°C $CT_{max}$. Due to low variance among individuals in 28°C $CT_{max}$, thermal plasticity was dependent on 12°C $CT_{max}$. Acclimation order (shape, circles = 12–28°C, triangles = 28–12°C) had a significant effect on $CT_{max}$ plasticity ($p \ll 0.0001$). Inset: plasticity in $CT_{max}$ has no significant correlation with 28°C $CT_{max}$ ($p = 0.71$). p-values from linear model; shaded region is 95% confidence interval, $N = 88$.

approximately 4°C but individuals acclimatized to 28°C having a range of only approximately 1.25°C. Individuals with low $CT_{max}$ at 12°C had greater acclimation response than individuals with high thermal tolerance at 12°C because there was little $CT_{max}$ variance at 28°C; thus, the acclimation response for thermal tolerance was significantly and inversely related to an individual's 12°C $CT_{max}$ ($R^2 = 0.90$, $p < 0.001$) and unrelated to their 28°C $CT_{max}$ ($R^2 < 0.0001$, $p = 0.92$, figure 3). Due to the significant effect of acclimation order on 12°C $CT_{max}$, acclimation order also explained a large and significant proportion of variation in $CT_{max}$ plasticity (38.58%, $p \ll 0.0001$).

## 3.4. Substrate-specific cardiac metabolic rate

For CaM, individuals were only assayed at one acclimation temperature of 12 or 28°C. CaM was measured using four separate substrates (glucose, fatty acids, lactate plus ketones plus ethanol, and endogenous) for all individuals. CaM for all substrates were significantly related to both body mass and heart mass ($R^2$ 0.22–0.44). Heart mass tended to have higher $R^2$ than body mass; however, because heart mass was not available for all individuals, body mass was used to correct for allometric scaling. Heart mass and body mass were significantly related ($R^2 = 0.19$ or 0.62 for 12 or 28°C, respectively; electronic supplementary material, figure S3).

For CaM END (endogenous (i.e. no added substrates)) at 28°C, the N. Ref was significantly lower than both the TE and S. Ref populations ($p < 0.05$). For all other substrate and temperature combinations, there was no significant difference among populations. For both 12 and 28°C, CaM GLU (glucose substrate) was significantly higher than with any other substrate. CaM FA (Fatty acids) and CaM LKA had the second highest CaM with no significant difference between CaM FA and LKA rates. CaM END at 12°C was not significantly different from LKA but was lower than all other substrates for both 12 and 28°C. For CaM END, individuals measured at 28°C had a significantly lower CaM than any other substrate–temperature combination (figure 4).

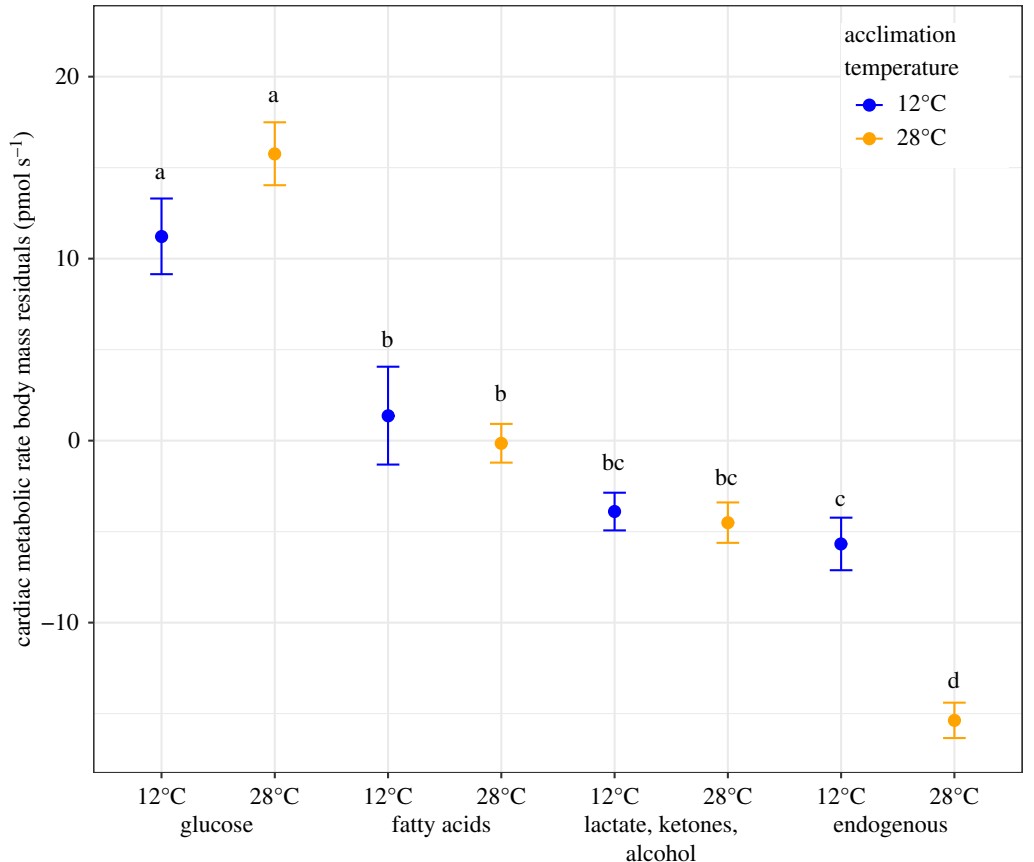

**Figure 4.** Cardiac metabolic rate substrate use. Blue 12°C, significantly higher use of glucose than fatty acids and lactate ketones alcohol (LKAs), significantly higher use of fatty acids than endogenous and no difference between endogenous and LKA CaM (ANOVA). Glucose $N = 58$, fatty acids $N = 50$, LKAs $N = 58$, endogenous $N = 55$. Orange 28°C, significantly higher use of glucose than all other substrates with no difference in fatty acid and LKA use, which were both significantly higher than endogenous CaM (ANOVA). Glucose $N = 51$, fatty acids $N = 49$, LKAs $N = 50$, endogenous $N = 49$. Mean ± s.e. Letters indicate significantly different groups across both temperatures.

Except for endogenous metabolism, CaM was unaffected by acclimation temperature (figure 5). Overall, only 0.47% of CaM variation was explained by temperature. Much more variance in cardiac metabolism was explained by substrate and body mass: substrate explained 29.1% of the variance, and body mass explained 16.3%. Across all exogenous substrates (i.e. without END), the average difference between CaM at 12 and 28°C was very small, only 2.75 ± 2.24 pmol oxygen s$^{-1}$ (= 7.6% of CaM at 12°C) (electronic supplementary material, table S2). This is in contrast with results from WAM and CT$_{max}$ where acclimation temperature had a significant and strong effect on physiological trait variation, explaining up to 96% of variation among individuals (for CT$_{max}$). As shown by the large confidence interval when comparing 12 and 28°C groups, CaM variation within an acclimation temperature among individuals was high (CV = 22–54% for mass-corrected CaM, table 1). Ventricular mass was significantly higher at 12°C than 28°C ($p < 0.001$, figure 5) despite no significant body mass difference between groups ($p > 0.40$).

When using heart mass rather than body mass to correct for allometric scaling (decreased sample size due to missing heart mass measures for four individuals), the relationship among substrate : temperature-specific CaM did not change. That is, there was still no significant acclimation response for CaM despite a significantly lower heart mass in individuals at 28°C (electronic supplementary material, figure S4).

## 3.5. Variation and covariance among individuals in complex metabolic phenotypes

All populations had high interindividual variation, which probably contributed to few significant differences among populations (e.g. 2.56-fold greater variation within than among population variance

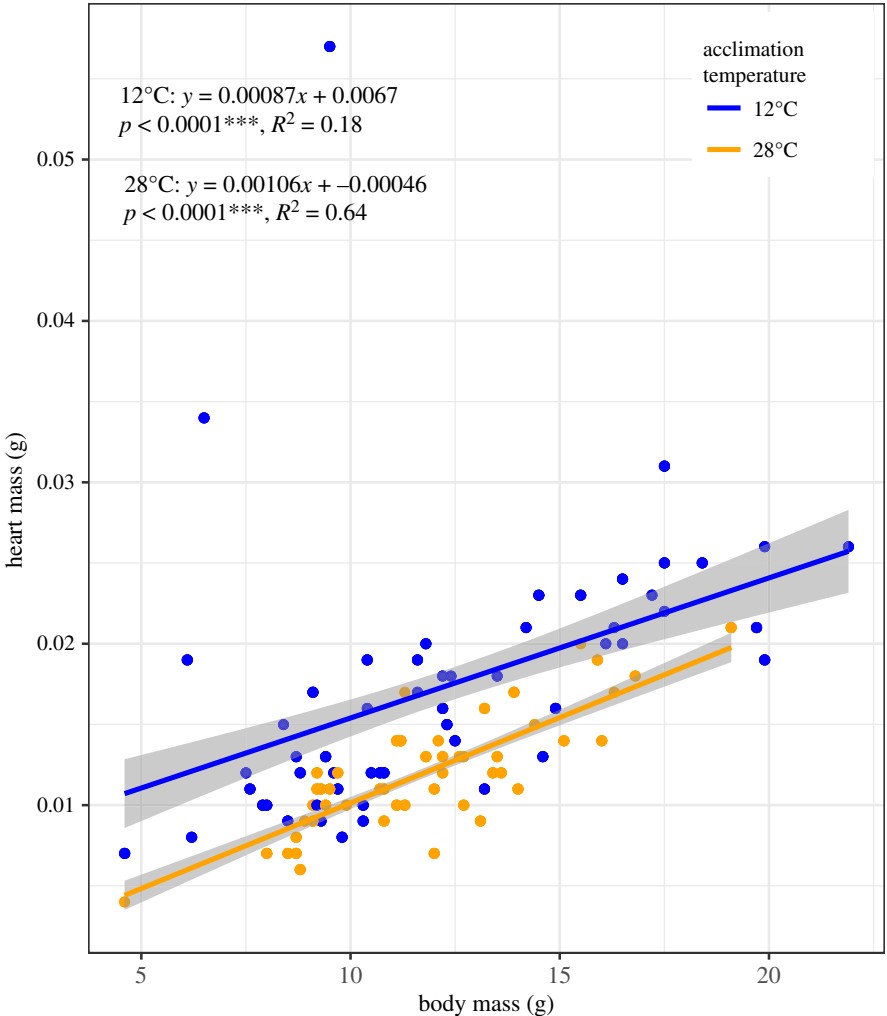

**Figure 5.** Relationship between heart mass and body mass. Relationship between heart mass (ventricular mass) and body mass for individuals at 12°C (blue) and 28°C (orange) acclimation conditions. $N = 55$ at 12°C, $N = 50$ at 28°C. Cardiosomatic index at 12°C was significantly greater than at 28°C despite no difference in body mass between acclimation temperatures. $p$-values from linear regressions; shaded region shows 95% confidence interval.

for WAM). Variation in all physiological traits was generally higher at 12°C than 28°C (table 1), and CaM and WAM were more variable traits (CaM CV = 48.6% and WaM 49.9%) than $CT_{max}$ (CV = 8.06%).

To examine the relationships among physiological traits, we calculated partial correlation coefficients for the six physiological traits (table 2; electronic supplementary material, figure S6): WAM, CaM for all four substrates and $CT_{max}$ for 12 and 28°C. Among metabolic rates at 12°C, WAM was significantly ($p <$ 0.02) correlated with CaM using FA, LKA and END substrates (table 2; electronic supplementary material, figure S6), with only CaM LKA having a positive correlation. Among substrate-specific CaM at 12°C, LKA was significantly and positively correlated with FA, GLU and END ($p < 0.03$). At 28°C, WAM was significantly and positively correlated ($p < 0.001$) with CaM using END substrate only. For CaM at 28°C, FA was significantly correlated ($p < 0.04$) with GLU (positive), LKA (negative) and END (positive), and LKA was significantly and positively correlated ($p < 0.005$) with GLU and END.

$CT_{max}$ at 12°C was significantly correlated ($p < 0.03$) with WAM and CAM. These correlations were negative except for $CT_{max}$ and CaM LKA. At 28°C, $CT_{max}$ was significantly ($p < 0.001$) and positively correlated with WAM and negatively correlated with CaM LKA and END.

## 4. Discussion

The results from this research suggest a large interindividual variation that could be important for the success of a species living in a highly temporally and spatially variable environment. Additionally, in

**Table 2.** Partial correlation coefficients. Partial correlation coefficients (above diagonal) and associated *p*-values (below diagonal) among physiological traits. All traits are residuals from mass-trait regressions. WAM (whole-animal metabolism) is $\log_{10}-\log_{10}$ regression. CaM (cardiac metabolism for FA, fatty acid; GLU, glucose; LKA, lactate–ketones–alcohol; END, endogenous) and $CT_{max}$ (critical thermal maximums) are linear regressions with mass. Italics are significant, bold with 'a' are significant with Bonferroni's correction.

| | WAM_Residuals | FA_Residuals | GLU_Residuals | LKA_Residuals | END_Residuals | $CT_{max}$_Residuals |
|---|---|---|---|---|---|---|
| **12°C partial correlations coefficients (above diagonal) and *p*-values (below diagonal)** | | | | | | |
| WAM_Residuals | — | *−0.2167* | −0.1372 | **0.3339**[a] | *−0.1828* | **−0.3393**[a] |
| FA_Residuals | *0.0189* | — | −0.1563 | *0.2038* | −0.1587 | *−0.2374* |
| GLU_Residuals | 0.1402 | 0.0925 | — | **0.6394**[a] | −0.0656 | *−0.2022* |
| LKA_Residuals | **0.0002**[a] | *0.0275* | **<0.0001**[a] | — | **0.3673**[a] | **0.4285**[a] |
| END_Residuals | *0.0485* | 0.0874 | 0.4819 | **<0.0001**[a] | — | **−0.3632**[a] |
| $CT_{max}$_Residuals | **0.0002**[a] | *0.0099* | 0.0288 | **<0.0001**[a] | **<0.0001**[a] | — |
| **28°C partial correlations coefficients (above diagonal) and *p*-values (below diagonal)** | | | | | | |
| WAM_Residuals | — | 0.1488 | 0.0071 | 0.0891 | **0.4171**[a] | **0.5463**[a] |
| FA_Residuals | 0.0978 | — | *0.2336* | −0.1797 | **0.4642**[a] | 0.1677 |
| GLU_Residuals | 0.9377 | *0.0087* | — | **0.3201**[a] | −0.1501 | −0.0083 |
| LKA_Residuals | 0.3229 | *0.0449* | **0.0003**[a] | — | **0.5398**[a] | **−0.3732**[a] |
| END_Residuals | **<0.0001**[a] | **<0.0001**[a] | 0.0947 | **<0.0001**[a] | — | **−0.3115**[a] |
| $CT_{max}$_Residuals | **<0.0001**[a] | 0.0616 | 0.927 | **<0.0001**[a] | **0.0004**[a] | — |

[a]Significant with Bonferroni's correction.

**Table 3.** Interindividual variance (CV) between acclimation temperatures. Column 3 is the ratio of CV at 12 and 28°C. Colour indicates high (red) or low (blue) CV at 12°C compared with 28°C.

| physiological trait | interindividual variance for 12 versus 28°C | ratio of CVs: 12/28°C |
|---|---|---|
| WAM | > | 1.23 |
| $CT_{max}$ | ≫ | 2.73 |
| CaM_GLU | > | 1.44 |
| CaM_FA | ≫ | 2.46 |
| CaM_LKA | = | 0.89 |
| CaM_END | = | 0.99 |

spite of the large interindividual variation, the variation in plasticity (acclimation response) was a function of physiological rates at 12°C. Finally, there were large differences in plasticity for the different traits: CaM were nearly the same when acclimatized and measured at 12 and 28°C, yet WAM had a 2.32-fold difference. The interpretation, including the effects of technical, biological and within-individual variation, is detailed below.

## 4.1. Interindividual variation

Interindividual variations for the six traits in *F. heteroclitus* are large (table 1) and tend to be larger at the lower acclimation temperature (table 3). Specifically, individuals acclimatized to 12°C tended to have greater interindividual variation (as measured by CV) than individuals acclimatized to 28°C. The largest difference in interindividual variance between acclimation temperatures was for $CT_{max}$; here, CV was 2.7-fold higher at 12°C than 28°C. For WAM, the CV was 1.2-fold higher at 12°C than 28°C. This consistent reduction in interindividual variation at 28°C for WAM and $CT_{max}$, measured in the

same set of individuals, suggests that acclimation to the higher temperature dampened interindividual variation. Similar reductions were seen for CaM FA and CaM GLU, where 12°C CVs were 2.5- and 1.4-fold higher (respectively) than 28°C CVs. This was not observed for CaM LKA or END, which had similar CVs at 12°C and 28°C acclimations (0.89-fold and 0.99-fold, respectively). One explanation for these data is that individual variation is suppressed with stress, which would suggest that for *F. heteroclitus*, 28°C is more stressful than 12°C. This idea is supported by evidence that maintaining *F. heteroclitus* at high temperatures (greater than 33°C) for extended periods of time results in poor survival [29,30].

Quantifying interindividual variation (variation among individuals) has allowed for variation in acclimation response within and among complex traits to be distinguished [49]. Not only is there a difference in the interindividual variances measured here due to acclimation, but for *Fundulus*, this variation is surprisingly large, and we suggest is biologically meaningful and not due to technical or random effects. For WAM using the same *F. heteroclitus* individuals, the variance among replicate measurements taken on different days was 7.5-fold lower that the variation among individuals [40]. This suggests that random biological variation is not the primary cause of the high interindividual variation. The technical variation [40] in our methods also is low because the variation in replicate measures measured in different chambers and on different days is small and the variation in measurements made overnight for each chamber is small (repeatability among days = 0.96 [40]). WAM was quantified using an HIFR [40], all individuals for the first acclimation condition were measured within one week of each other, and all measures across acclimation conditions occurred within six weeks. This short time period should reduce the variation among individuals due to time (changes in size or other variables correlated to time). Similarly, for CaM, methodological validation of the high-throughput micro-respirometer used to measure tissue-specific oxygen consumption demonstrated low technical variation ($O_2$ flux <1% of CaM for all chambers [28]). CaM for all individuals were measured within three weeks, reducing variation due to time despite the large sample size.

The magnitude of the interindividual variation (CV) within an acclimation temperature for WAM was 36% at 12°C and 30% at 28°C, while CaM CV for FA was 54% at 12°C and 54% at 28°C. In comparison to WAM and CaM, $CT_{max}$ had little interindividual variation (less than 3% and less than 1% for 12°C and 28°C, respectively). This interindividual variation for metabolic rates exceeds values previously reported for other teleost fish [50,51]. For example, in brown trout variation among individuals in SMR and maximum metabolic rate ranged from 12 to 14% [50]. If this variation is heritable as suggested by other studies [51–53] and the individual variation in metabolic rates is associated with differential survival and other life-history traits [35,54–57]; the greater variation among individuals may be important for evolutionary adaptation.

While the data here cannot address whether the interindividual variation within an acclimation condition reflects a heritable, evolutionary important trait, previously we have suggested that this variation reflects heritable changes that are evolutionarily important [49]. This assertion was based on the association between CaM and mRNA expression where up to 81% of the interindividual variation in substrate-specific CaM was explained by metabolic gene expression variation [34]. Substrate use variation among individuals was explained by gene expression variation among different metabolic pathways (expression of glycolytic enzymes, oxidative phosphorylation enzymes or Krebs cycle enzymes) [34]. Additionally, a separate study using *F. heteroclitus* found lower gene expression variation among populations with reduced heterozygosity compared with wild-caught individuals [58]. Similarly, cardiac mitochondrial metabolism was associated with the DNA variation in both nuclear and mitochondrial genotypes [59,60]. These data suggest that there is a genetic basis for the individual variation in the metabolic traits measured here. Assuming that the heritability of interindividual variation is significant and impacts fitness, the magnitude of variation in a trait may be explained if there are multiple optimum environments or if there are multiple combinations of physiological traits that are advantageous as suggested by prior data on CaM and mRNA expression [34,60]. This hypothesis will require further investigation to identify if genetic variation is associated with the interindividual variation in metabolic traits.

## 4.2. Evidence for local adaptation

Dayan *et al.* [32] found evidence of genotypic divergence between the TE and two reference populations (different from the reference populations used here) and a significant difference in $CT_{max}$ after 28°C acclimation in individuals collected from the TE site and the north reference (Mantoloking, NJ, 40°

3′0.02″ N, approx. 25 km north of the N. Ref site used here, 39°52′28.0″ N [32]). By contrast, Healy *et al.* [61] found that New Jersey *F. heteroclitus* populations between latitudes of approximately 39° and approximately 40° had little difference in $CT_{max}$ when acclimatized to 15°C despite diverging mitochondrial genotypes. Yet, the different acclimation temperatures used in these two studies may account for the apparently contrasting evidence for or against a steep cline in $CT_{max}$ along the New Jersey coast for *F. heteroclitus*. We found that under cooler (12°C) acclimation conditions, interindividual variation was greater than under warmer (28°C) acclimation conditions, which could explain why a set of populations at similar latitudes did not differ in $CT_{max}$ when measured at 15°C [61] but did differ at 28°C [32]. This suggests that that there may be a steep change in $CT_{max}$ along the northern New Jersey coast between 39° and 40° of latitude, which is only measurable under warm (greater than 15°C) acclimation conditions.

Despite this prior evidence of local adaptation in the TE population, there was a surprising lack of physiological trait divergence among the populations used here (only CaM END at 28°C was different among populations). This could be due to connectivity (migration) among populations where the sharing of allelic variants inhibits any local evolutionary response, a lack of sufficiently strong selective force among populations, the traits measured are difficult to evolve or not selectively important, or that maintenance of variation in these traits is more important than canalization towards an optimum trait value. While migration among populations is possible, as populations are 10 or less kilometres apart, prior studies suggest a small home range (100s of metres) for this species [62,63]. Additionally, polluted *F. heteroclitus* populations on a similar geographical scale have adaptive genetic divergences [33,64,65]. Similarly, adaptive genetic divergence can exist within *F. heteroclitus* populations between mitochondrial haplotypes [60] or between microhabitats [66,67]. Thus, migration *per se* has not previously inhibited adaptive divergence among well-connected *F. heteroclitus* populations exposed to different environments. It is possible that the 4°C summer temperature difference (figure 1) is not a sufficiently strong selective force. Yet, this argument too is not supported by clinal variation among populations north of the Hudson river, where the 4°C difference is related to clinal genome-wide changes (Dayan *et al.* 2020, unpublished data). This leaves two, more complex, reasons for the lack of divergence among physiological traits: (i) these traits are either difficult to evolve (e.g. due to antagonistic pleiotropy) or are not selectively important in this TE environment or (ii) that maintenance of variation in these traits is more important than canalization towards an optimum trait value. Additionally, interactions between ecological and evolutionary dynamics in response to temperature variation may result in similar phenotypic optima despite apparently different environments obscuring local adaptation patterns [68,69]. This could include epigenetic effects if variation in a trait can be attributed to epigenetic changes within a generation (e.g. DNA methylation) as well as heritable genetic and epigenetic changes across generations (e.g. single nucleotide polymorphisms, maternal effects) [70,71]. To examine epigenetic effects, DNA methylation patterns (for example) in genetically similar individuals that also show divergence in physiological traits could be compared. To address the unexpected lack of divergence among populations, genes associated with the high interindividual variance could be identified and patterns in polymorphisms in these genes partitioned among populations. DNA methylation data in combination with genetic data could also be used to identify single nucleotide polymorphisms important for explaining variation in methylation patterns important for physiological trait variation (i.e. meQLT analysis [71]).

## 4.3. Correlations among physiological traits

The large number of individuals measured here provided the opportunity to address relationships among traits using partial correlations within each acclimation temperature (table 2; electronic supplementary material, figure S6). At 12°C, CaM LKA was uniquely positively correlated with all other traits. All other CaM substrates (FA, GLU and END) were negatively correlated with $CT_{max}$ and WAM with no significant correlations among CaM FA, GLU and END. While correlation does not show causation, these data suggest a quantitatively different effect of CaM LKA on whole-animal phenotypes when compared with FA or GLU and suggests that metabolism of secondary metabolites (lactate, ketones and alcohol) is biologically different.

At 28°C, probably due to lower interindividual variation, there were fewer significant partial correlations. Unlike at 12°C, $CT_{max}$ was positively correlated with WAM, which could indicate a metabolic cost of thermal tolerance under 28°C acclimation conditions. All CaM substrates were significantly correlated with the exception of GLU and END. WAM was positively correlated with END CaM only. This may reflect a higher reliance on GLU at 28°C with 92% of individuals (47 of 51)

having the highest CaM with GLU substrate (compared with 64% at 12°C; electronic supplementary material, figure S5).

The physiological traits measured here may be biologically important and evolving independently. Alternatively, these traits might be affected by a similar suite of genes due to pleiotropy. Regardless, the observation that correlation between traits is temperature dependent suggests that the genetic basis for these correlations may be temperature dependent. Previously, mRNAs with adaptive quantitative expression difference among populations were unrelated at 12, 20 and 28°C acclimation temperatures [8]. Although it is speculative until genomic analyses are applied to these data, the correlations suggest that the genes of importance, that explain the physiological trait variation will be shared among traits at a given acclimation temperature but will be different at different acclimation temperatures. Finally, although there were significant correlations among substrate-specific CaM rates, there was interindividual variation in substrate use. That is, although at 12°C there was a positive correlation between LKA and all other substrates, a given individual may have relatively high use of one substrate and relatively low use of another. The same was true at 28°C where individuals differ in their relative substrate use (electronic supplementary material, figure S5).

Few previous studies have investigated relationships among complex physiological traits in the same set of individuals measured after acclimation to different temperatures as presented here. However, Nyboer & Chapman [72] found a significant negative relationship between the percentage of compact myocardium and SMR and a significant positive relationship between both ventricular mass and $CT_{max}$ and ventricular mass and aerobic scope. This demonstrates that cardiac remodeling can influence whole-animal physiological traits including WAM rate and $CT_{max}$. Tissue-specific (heart) oxygen consumption rates, although not substrate-specific, also have similar allometric scaling as WAM rates in several teleost fishes (including *F. heteroclitus*), indicating that heart oxygen consumption may be predictive of whole-animal oxygen demands [73]. We demonstrate that WAM and CaM are correlated for some temperature–substrate combinations, potentially identifying biologically important differences in metabolite use under different temperature conditions. Additionally, we find evidence of cardiac remodeling (significant difference in heart mass between acclimation temperatures despite no difference in body mass, figure 5) that may allow individuals acclimatized to different temperatures to maintain cardiac output, which is important for tissue oxygen delivery and WAM processes maintenance.

## 4.4. Acclimation order

The order of acclimation (12°C followed by 28°C or 28° followed by 12°C) significantly affected WAM and $CT_{max}$ measured at 12°C. For WAM, acclimation order and the interaction between acclimation order and acclimation temperature were significant (electronic supplementary material, table S1): individuals acclimatized to 12°C first had significantly higher WAM at 12°C than individuals acclimatized to 28°C first. Similar to WAM, $CT_{max}$, acclimation order and the interaction between acclimation order and acclimation temperature were significant (electronic supplementary material, table S1 and figures S1 and S2). However, there was an opposite effect: there was a significantly higher $CT_{max}$ at 12°C (average approx. 1°C) for individuals acclimatized to 28°C first. Thus, acclimation to 28°C had a residual effect on WAM and $CT_{max}$ when acclimatized to 12°C. A simple explanation for these data is that acclimation to higher temperatures is more effective than acclimation to lower temperatures, such that the effect of higher temperatures has a hardening effect that is not readily removed by low acclimation and that acclimating to lower temperatures from high temperatures takes longer.

Healy & Schulte [74] demonstrated that repeated measures in *F. heteroclitus* had no effect on $CT_{max}$ when acclimatized to 15°C, but that three weeks of acclimation may not be sufficient when going from warm to cold temperatures. However, in zebrafish, $CT_{max}$ increased after an initial $CT_{max}$ determination (repeated measures 7 days apart) suggesting a residual effect [24]. That is, the first exposure 'hardened' individuals, enhancing their thermal tolerance upon repeated measure [24]. This seems reasonable, because determination of $CT_{max}$ requires exposing individuals to nearly lethal temperatures. Yet, results presented here are different in that hardening was only observed when acclimatized to 28°C first: there was no difference in 28°C $CT_{max}$ between acclimation groups (12 or 28°C first). We suggest that while typical acclimation responses may be relatively quick in *F. heteroclitus*, exposure to high, nearly lethal temperatures can have a lasting effect when acclimatized to a high temperature. Thus, it may be that the combination of high acclimation temperature and the higher $CT_{max}$ temperature (average $CT_{max}$ 42.5°C at 28°C acclimation) has a lasting effect that is not

seen at low acclimation temperature (average $CT_{max}$ 36.3°C at 12°C acclimation). Therefore, higher absolute temperature exposure has a long-lasting residual effect that is not seen when acclimatized to low temperature. Interestingly, these same individuals acclimatized to 28°C first had a significantly lower WAM at 12°C than individuals acclimatized to 12°C first. Thus, for both $CT_{max}$ and WAM, 12°C performance was affected by previous thermal conditions while 28°C performance was not. This suggests that acclimation to a higher temperature is more effective because performance at the higher acclimation temperature was independent of prior thermal experience.

A corollary to acclimation to higher temperature being more effective is that the four-week acclimation to 12°C from 28°C was insufficient. Prior studies in *F. heteroclitus* found that acclimation from 15°C to 5°C or 15°C to 25°C occurred within three weeks and that acclimation to cooler temperatures occurred more slowly [74]. This would explain the significant effect of acclimation order on plasticity in $CT_{max}$ (electronic supplementary material, figure S2), which results from a higher 12°C $CT_{max}$ if acclimatized to 28°C first (acclimatized to 28°C before acclimation to 12°C). Yet, within a single acclimation order group, there was still substantial variation in the degree of acclimation response (CV = 9.69% for 12–28°C, CV = 13.35% for 28–12°C) and importantly, plasticity is still a function of the 12°C response within each acclimation order group ($R^2 = 0.89$ for 12–28°C, $R^2 = 0.87$ for 28–12°C, $p \ll 0.0001$ for both). The similar CV for plasticity in $CT_{max}$ between acclimation groups, as well as the similar significant slope between 12°C $CT_{max}$ versus *plasticity*, suggests that this variation exists when acclimating from both warm to cool and cool to warm temperatures. While it is unlikely that the high and nearly lethal temperatures experienced during $CT_{max}$ would occur, the acclimation temperatures used here are ecologically relevant and thus variation in acclimation response within an acclimation order group represents biologically relevant interindividual variation.

## 4.5. Thermal performance acclimation effects and adaptation

The expectation for metabolic rates is that an active physiological acclimation response (versus the passive effect of temperature on physiological performance) will result in lower rates than the acute effect of temperature [75,76]. For this study, a passive temperature effect is expected to be 3.0- to 4.4-fold increase in metabolism for the 12–28°C temperature measured here (i.e. $Q_{10} = 2.0$–2.5, [75,77,78]). This large change predicted from a passive response to temperature does not occur for the *F. heteroclitus* metabolic rates (table 1). Thus, responses to 12 and 28°C most likely represent an active acclimation effect (but see discussion below). However, acclimation effects were surprisingly different among WAM and substrate-specific CaM (table 1). Overall, 28 versus 12°C had a minimum effect on CaM (0.65- to 1.1-fold change, $Q_{10} = 0.76$–1.07, table 1), yet resulted in a 2.32-fold increase in WAM ($Q_{10}$ of 1.68). Additionally, and more importantly, the acclimation response varied among individuals, where individuals with the largest increases in WAM between 12 and 28°C had lower rates at 12°C (figure 2). For $CT_{max}$, a greater acclimation response to 28°C was also associated with individuals with lower 12°C $CT_{max}$.

For substrate-specific CaM, there was no significant difference in 12 and 28°C metabolic rates for exogenous fuel (glucose, fatty acids and LKA), but there was a reduction for endogenous metabolism (END). For CaM END, the significant decrease in metabolism at 28°C relative to 12°C ($Q_{10} = 0.76$) resulted in 1.55-fold higher CaM at 12°C than 28°C ($p = 0.050$, figure 4; electronic supplementary material, table S2). This lack of difference between temperatures in CaM for exogenous substrates suggests that for CaM, individuals have a nearly perfect acclimation response. Alternatively, acclimation may have no effect and instead the response is similar to an acute response with a single thermal performance curve (TPC) and a sharp peak between the two acclimation temperatures (e.g. 20°C). In this scenario, CaM measured at 12 and 28°C fall on either side of the peak of the TPC and thus have equivalent rates [78]. Thus, without characterizing the TPC for substrate-specific CaM at both 12 and 28°C, the acclimation response cannot be distinguished with certainty. Yet, among *F. heteroclitus*' TPCs for muscle contraction, cardiac and liver mitochondrial metabolism, and hatching time, there is no case where the thermal optimum peak is lower than 28°C, nor do they have sharp peaks with equivalent physiological rates among temperatures commonly experienced by *F. heteroclitus* [59,79–81]. These studies suggest that the low $Q_{10}$ for CaM measured at 12 and 28°C is an active acclimation response.

One additional observation presented here supports a strong compensatory acclimation response for CaM: the significantly larger ventricular mass independent of body mass in individuals acclimatized to 12°C versus 28°C (ANCOVA, ventricular mass, acclimation temperature $p < 0.001$; electronic supplementary material, figure S4). The larger ventricular mass at 12°C is suggestive of cardiac remodeling that could contribute to temperature compensation. This might also explain the reduction of END cardiac metabolism at 28°C because smaller hearts would be expected to have less

endogenous reserves. Similar patterns in ventricular mass have been observed in *F. heteroclitus* [79] and other fish species and have been the result of changes in relative thickness of compact and spongy myocardium [72,82–84]. Cardiac remodeling in response to cold acclimation could allow for the maintenance of cardiac output by increasing muscle mass, which increases stroke volume to compensate for decreased heart rate at lower temperatures [83]. Additionally, the proportion of compact myocardium, positively correlated with ventricular mass, has been correlated with increased SMR and aerobic scope [72]. Independent of body mass, heart mass is significantly correlated with each substrate-specific CaM at both acclimation temperatures. This suggests that larger heart mass can increase CaM and may explain why there is no significant difference in CaM despite a 16°C difference in measurement temperature. Thus, the nearly equal CaMs for all exogenous fuels when acclimatized and assayed at 12 and 28°C with a compensatory change in ventricular mass is most likely indicative of a strong physiological acclimatory response that compensates for the 16°C difference in environmental temperature.

Data supporting a compensatory acclimation response in CaM are in contrast with WAM data where there were significantly higher rates at 28°C than 12°C (2.32-fold, table 1). This 2.32-fold increase with acclimation is less than the 3- to 4.4-fold increase expected for a passive response to a 16°C temperature change [75,85], and the contrast to CaM suggests that acclimation is active but less effective for WAM. Similar results for WAM in longjaw mudsucker (*Gillichthys mirabilis*) acclimatized at 19 and 26°C demonstrate a lack of active acclimation to temperature with a $Q_{10}$ of 1.6 and 2.7 between ambient (13–14°C) and 19 or 26°C, respectively, while there was perfect acclimation to 9°C (no difference from at ambient) [86]. Other data also suggest a limited ability to compensate metabolic rate through acclimation in many fish species ([29,87–89], although see [90–92]), including in response to cold acclimation in *F. heteroclitus* [88]. This contrasts with the expectation that maintenance of lower oxygen consumption at higher temperatures via acclimation would improve fitness [93]. Thus, it is not clear why the acclimation response for WAM is relatively low; it may simply be physiologically difficult to evolve or have other fitness cost. Yet, in sheepshead minnow, an ecologically similar estuarine species, thermal tolerance is gained asymmetrically during acclimation to warm or cold temperatures (gain 50% of thermal tolerance when acclimating from approx. 11°C to approx. 18°C, [94]). That is, exposure to a different temperature, regardless of the magnitude of increase or decrease in temperature, results in a gain of the majority of physiologically available thermal tolerance (or loss for cold acclimation) within approximately 20 days. If a similar mechanism for thermal tolerance is present in *F. heteroclitus*, our evidence of incomplete active acclimation may represent the majority of thermal tolerance accruement available for this species.

$CT_{max}$, which is an endpoint assay of an acute response, is significantly higher at 28°C than 12°C. This increase in $CT_{max}$ at higher acclimation temperature is thought to be advantageous because this leads to a higher maximum survival temperature [24,36,95]. Higher $CT_{max}$ with higher acclimation temperatures is common in teleost fish [72,74,96–99]. What was interesting and novel in the data presented here was that the acclimation response (figure 4) was almost wholly dependent on the $CT_{max}$ at 12°C ($R^2 = 0.93$, $p < 0.001$) with little variation in the acclimation response explained by $CT_{max}$ at 28°C ($R^2 < 0.001$, $p > 0.5$). The observation that 93% of the variation in $CT_{max}$ plasticity is explained by 12°C $CT_{max}$ indicates that there is little unaccounted within-individual variation. The lack of significant $R^2$ for 28°C $CT_{max}$ could arise from a large random error or unaccounted for within-individual variation (e.g. time of day, or unaccounted for stress factors). However, this is unlikely because there is little variation in 28°C $CT_{max}$ (tables 1 and 3) and these 28°C $CT_{max}$ measures exist in plasticity (log$_2$ ratio 28/12), which is accounted for by 12°C $CT_{max}$ (figure 3). Thus, we suggest that the reason plasticity is unrelated to 28°C $CT_{max}$ is not due to random error or unaccounted within-individual variation but instead reflects small variation when approaching the upper limit of thermal tolerance at higher temperatures.

The observation that plasticity is mostly explained by 12°C $CT_{max}$ implies that individuals with a lower $CT_{max}$ at 12°C (within each acclimation order group) had a greater acclimation response (figure 3). These results are similar to those in Morgan *et al.* [24] where zebrafish with an initially low $CT_{max}$ had a greater difference between $CT_{max}$ in repeated trials. If a larger active physiological response is adaptive, then the greater acclimation response for $CT_{max}$ comes at a price of having a lower tolerance to extreme temperature when acclimatized to low temperature and may also be due to variation in time it takes to acclimate. It is likely that in *F. heteroclitus*, which experience a temperature range from below freezing in winter to above 30°C during the warmest summer months in the southern range, thermal performance and time to acclimation at both temperatures is ecologically relevant.

Finally, there is a similar pattern of variation for acclimation response among individuals for WAM and $CT_{max}$ where low 12°C performance is associated with greater acclimation response (figure 2). However, unlike for $CT_{max}$, plasticity in the WAM acclimation response was a function of both 12 and 28°C WAM (figure 2*a*,*b*). Thus, individuals with low 12°C WAM and high 28°C WAM have greater plasticity than individuals with high 12°C WAM and low 28°C WAM. This response, where plasticity is a function of both 12°C WAM and 28°C WAM is for all 58 individuals with WAM measured at both acclimation temperatures and is similar for all three plasticity groups (figure 2*c*). Unaccounted within-individual variation would affect the $R^2$ (part of the error term of the linear model; therefore, repeated measures of individuals could make the relationship between WAM and plasticity more precise.

While there was a significant relationship between plasticity and both 12 and 28°C WAM, surprisingly, there was no overall significant correlation between performance at 12 and 28°C for WAM until individuals were grouped by the magnitude of acclimation responses (lower 10%, middle and upper 10% confidence interval, or CI, figure 2*c*). Interestingly, the strength of the relationship between performance at 12 and 28°C also differed, with a significant correlation only in the middle group (between lower and upper 10% CI, $R^2 = 0.30$); little correlation occurred for the lower 10% CI ($R^2 = 0.04$) and a negative correlation occurred for the upper 10% CI ($R^2 = 0.49$). The significant relationship between 12°C WAM and 28°C WAM is only found among individuals with moderate plasticity and not among individuals with extreme high or low plasticity. This could reflect differences in acclimation response: individuals with very low plasticity (green figure 2*c*,*d*) tended to have low 28°C WAM and individuals with very high plasticity (red figure 2*c*,*d*) tended to have low 12°C WAM. Thus, the potential difference among individuals in acclimation response affects the relationship between 12 and 28°C WAM.

Alternatively, unaccounted within-individual variation may explain why there is no significant relationship between 12°C WAM and 28°C WAM in individuals with extreme plasticity. Specifically, measures of WAM at one temperature for individuals with very low or very high plasticity (upper and lower 10% CI) were impacted by some individual biological effect (e.g. stress response). Thus, for these individuals with extreme plasticity, there may have been a lack of repeatability in WAM measurement due to biological and not technical variation impacting the ability to assess acclimation response and correlation between performance at 12 and 28°C. However, these data are similar to $CT_{max}$ and suggest that variation in WAM performance at 12°C affects variation among individuals in acclimation response. These results, where acclimation response is a function of metabolic rate are consistent with a prior study using a coral reef fish, *Lates calcarifer*, where individuals with an initially high metabolic rate acclimation to cooler condition had a smaller acclimation response than individuals with an initially low metabolic rate [100]. As previously discussed, this may be due to variation among individuals in the ability to acclimate, especially when going from 28 to 12°C acclimation conditions. Alternatively, a trade-off between temperature-specific responses and the magnitude of acclimation responses could explain the correlation between low-temperature performance and magnitude of acclimation response.

The intra-individual variation in physiological performance and the acclimation response may be evolutionarily important if it is heritable. $CT_{max}$ has a significant heritability and has been readily selected for in wild-caught zebrafish over six generations [37]. For cold hardening in *Drosophila melanogaster*, there is a strong heritability and significant differences among individuals that affect acclimation response [36]. In two species of damsel flies, variation in thermal response affects both reproduction success and survival [22]. Similarly, metabolic rates have high heritability [51–53], and the individual variation in metabolic rates is associated with differential survival and other life-history traits [35,54–57]. Finally, across taxa, there is a strong phylogenic signal for WAM and biogeographic distribution and habitat use [55,101]. These data indicate that the variation in both $CT_{max}$ and WAM, as well as the interindividual variation in acclimation response are evolutionarily important. While we cannot determine, without a direct fitness measure, if the interindividual variation in this study is advantageous, we suggest that living in a highly variable environment could favour the maintenance of interindividual variation in both physiological performance at specific temperature and the magnitude of acclimation response.

# 5. Conclusion

By observing variation in multiple physiological traits among more than 100 individuals, we found that (i) high variation among individuals within populations exceeded variation among populations, (ii) high

variation in physiological performance at 12°C acclimation explained variation in acclimation response for WAM and $CT_{max}$ such that individuals with low performance at 12°C also had the greatest acclimation response, yet, (iii) individual trait performance was correlated among traits and these correlations were acclimation temperature specific. The large sample size and diverse phenotypes used here provide a strong foundation for future investigations of genotypic and phenotypic variation among these populations. Specifically, a future investigation of mRNA expression patterns in metabolically active tissues collected from these individuals along with genotype–phenotype analysis would further our understanding of how wild populations respond to local temperature variation. Additionally, TPCs for substrate-specific CaM and cardiac output in addition to CaM would provide evidence of physiological compensation or demonstrate a shift in the TPC as a result of temperature acclimation. Overall, our findings suggest interindividual variation in trait-specific physiological responses to temperature acclimation. We find that some individuals may have a greater capacity for acclimation response than others and demonstrate a need for additional research investigating interindividual variation in physiological plasticity of complex traits, which are relevant for global climate change response in many species.

Ethics. All animal use was approved by the University of Miami Institutional Animal Care and Use Committee.

Data accessibility. All data are available at the Dryad Digital Repository (https://doi.org/10.5061/dryad.0gb5mkm0w) [102]. All scripts used in analysis can be found on github: https://github.com/mxd1288/physiological_plasticity_funhe and have been archived with the Zenodo repository: https://doi.org/10.5281/zenodo.4913597.

The data are provided in electronic supplementary material [103].

Authors' contributions. M.K.D., A.N.D., M.A.E., D.L.C. and M.F.O.: experimental design. M.K.D., A.N.D. and M.A.E.: data collection. M.K.D., D.L.C. and M.F.O.: data analysis and visualization. M.K.D., D.L.C. and M.F.O.: writing of the manuscript. Funding obtained by D.L.C. and M.F.O.

Competing interests. No conflicts of interest are declared.

Funding. Funding from National Science Foundation award nos. IOS 1556396 and IOS 1754437.

Acknowledgements. Thank you to Dr Chris Langdon for advice and assistance in high-throughput equipment design and to Liam Dorsey, Agatha Freedberg and Rebecca Vanarnam for work on data collection and fish husbandry.

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
