## [Peer Review File · Royal Society Open Science]

Review History

RSOS-210440.R0 (Original submission)

Review form: Reviewer 1 (Patricia Schulte)

Is the manuscript scientifically sound in its present form?

Yes

Are the interpretations and conclusions justified by the results?

Yes

Is the language acceptable?

Yes

Do you have any ethical concerns with this paper?

No

Have you any concerns about statistical analyses in this paper?

No

Recommendation?

Accept with minor revision (please list in comments)

Comments to the Author(s)**General Comments**

This manuscript reports on an impressive dataset cataloging the extent of inter-individual variation in the acclimation responses in a variety of physiological traits in Atlantic killifish. Although between-population variation in plasticity of physiological traits has been described in a variety of species, inter-individual variation within populations has rarely been characterized, particularly in fish (although see Morgan, Finnogen and Jutfelt 2018), and thus this paper represents a significant advance in our understanding of this important topic.

The particular strengths of the study are the large sample size, the fact that multiple traits were measured for each individual (allowing for correlations among traits to be examined), and the high quality of the methods and statistical approaches used to assess the traits.

I found the paper to be generally clear and easy to follow.

I have only one major comment for the authors to consider. It does not really affect the integrity of the data, but it does suggest some potentially different interpretations. Specifically, I am concerned about the patterns apparent in the data presented in Supplemental Figures 1 and 2, which show that both metabolic rate and CTMax differ depending on the order of acclimation.

For example, CTMax at 12C is higher in individuals that were previously exposed to 28C acclimation. [Note that because of the design, the fish were initially brought through a pseudo winter at 10C, so fish that were acclimated to and tested at 12C first only underwent acclimation from 10 to 12C during the four-week acclimation period, whereas the fish that were first acclimated and tested at 28C and then acclimated to 12C had to undergo acclimation down from 28C to 12C in four weeks.]

In the supplemental results section, the authors suggest that the observed acclimation order effect is due to prior heat hardening because those fish were also exposed to a CTMax trial at the 28C acclimation temperature. However, there is another plausible explanation – that the fish that underwent acclimation to 12C from 28C were not yet fully acclimated to the lower temperature at the time of measurement.

Similar acclimation order effects are evident for metabolic rate, and these are unlikely to be explained by heat hardening due to the prior CTMax trial.

The authors consider and dismiss the possibility of incomplete low temperature acclimation in the supplemental results section based on data presented in Healy and Schulte (2012). However, the data in Healy and Schulte (2012) suggest that at 3 weeks acclimation of CTMax is not yet complete when the direction of acclimation is from high to low temperature and they do not report data on acclimations beyond this time. In general, in fish, acclimation to low temperatures is much slower than is acclimation to higher temperatures, and therefore this is an important consideration. As a result, I would hesitate to entirely dismiss the possibility that a significant fraction of the intraspecific variation in acclimation capacity that has been observed in this study may be the result of trial order. This idea is supported by the fact that trial order is a significant factor in the linear models.

I worry about the extent to which the patterns shown in Figures 2A and Figure 3 are being influenced by this trial order effect. If I look at the spread in the CTMax data, for example, it seems likely that the majority of the individuals with high CTMax at 12C (and thus low plasticity) are from the group that experienced 28-12C and vice versa. Indeed, the linear models reveal a strong effect of acclimation order on the models for whole animal metabolic rate, CTMax, and Cardiac metabolic rate. It is possible that this result is still robust, but it is difficult to know from the data as presented. It might be interesting to show this in the figures either using different shaped dots or different colours for the different trial orders. This would go a long way towards reassuring the reader of the origins of the observed patterns. Note that it does not change the pattern. But it might alter how you would interpret it.

Another interesting idea that follows from the possibility that low temperature acclimation might not be complete in the fish that were tested first at 28 and then at 12 is that individuals might vary in the rate of acclimation. That is, what if some individuals are not yet fully acclimated, while others have achieved complete acclimation? Thus, the authors might consider whether their observed results are consistent with differences in the capacity for acclimation (as they suggest), or might instead reflect inter-individual differences in the rate of acclimation to low temperature.

Whatever the case, I think it is absolutely critical that the issue of the effects of acclimation order be brought forward more clearly in the main text and discussed, and their implications for the conclusions drawn be fully considered. Otherwise, a casual reader of the paper who does not delve deeply into the supplements may miss the importance of these data. For example, they do not appear in the abstract.

Finally, a less interesting, less likely, but also possible, explanation for the observed patterns could stem from intra-individual variation in the traits that is not being captured that might influence the assessment of plasticity. Each trait is measured only once on each individual at each acclimation temperature, so the repeatability of the trait is not assessed here. How can you dismiss the possibility that variation in the plastic responses could be due to an inaccurate measure for an individual at one acclimation temperature and then regression/reversion to the mean at the other temperature? For example, imagine that (for whatever reason) the one measure of CTMax was unusually high for an individual acclimated at 12C (i.e. subsequent assessment of the trait in the same individual would have resulted in a higher value for CTMax at 12). If the assessments for this individual were close to the group mean at 28C, then you would get a low value for plasticity that might not actually be real. Although CTMax is thought to be pretty repeatable, the repeatability estimates are only in the range of 0.28-0.65 (Morgan et al 2018). If you modeled the data with this level of "error" in the estimates, how much would that change your relationships? On the other hand, Morgan et al (2018) found similar patterns where individuals with high innate tolerance have smaller acclimation responses, which helps to bolster the argument that this is a real pattern, rather than a reflection of error in the data (although regression towards the mean could be a problem in their data as well). For CTMax, if random variation is greater at 12C than at 28C acclimation, this would also play a role.

I think the potential for random error should be acknowledged a little more in the paper. Technical variation is considered and dismissed in several places, but biological variation isn't really discussed. I agree with the authors that their methods are outstanding and thus the likelihood of technical variation being the cause of the differences are low, but I am less sanguine about dismissing random biological factors given the known variation (moderate level of repeatability) in assessments of traits such as CTMax.

In addition to this major comment above, I have some minor suggestions for revision, outlined below, that occurred to me as I read the paper.

Line 1: I think you should consider altering the title. Although your initial design was intended to examine among-population variation in plasticity, that's not what you found. Instead, you found high levels of within-population (inter-individual) variation in plasticity. This inter-individual variation in plasticity is really interesting and is an important result, and I think it should be reflected in the title of the paper.

Line 17: The abbreviations for whole animal metabolic rate, critical thermal maximum and cardiac metabolic rate should be provided in brackets after each of the terms. Otherwise, the rest of the abstract is a bit difficult to follow.

Line 19: I know that it is impossible to fully explain your sample sizes, as they varied across traits and acclimation temperatures, but when I look at (for example) CTMax in the raw data, it looks like you only have matched 12 and 28C data for about 86 fish (about half the number reported in the abstract) and it is this matched data that is needed to compute individual plasticity. I haven't checked for the other traits but it looks similar for whole animal metabolic rate. I think a bit more transparency here would be helpful on this issue. I should note that this is still an extremely impressive sample size (even when divided across three populations).

Line 44: You emphasize the daily variation in the habitat of these fish here, but you are looking at plasticity in response to 4-week acclimation to a constant temperature. I think it is also important to also describe the seasonal variation in temperature in these habitats, since the type of plasticity you are looking at is probably more relevant to seasonal acclimatization

Line 66-68: Although I agree that in general interindividual variation in plasticity has received limited attention in animals, it has actually been looked at a fair bit in plants. As a result, I would maybe soften this statement a bit.

See for example:

van Rooijen, R., Aarts, M. G., & Harbinson, J. (2015). Natural genetic variation for acclimation of photosynthetic light use efficiency to growth irradiance in *Arabidopsis*. *Plant Physiology*, 167(4), 1412-1429.

Le, M. Q., Engelsberger, W. R., & Hincha, D. K. (2008). Natural genetic variation in acclimation capacity at sub-zero temperatures after cold acclimation at 4 C in different *Arabidopsis thaliana* accessions. *Cryobiology*, 57(2), 104-112.

And there is the paper by Morgan et al (2018) which you cite elsewhere in the paper that looked at interindividual variation in acclimation of CTMax in fish, which should probably be mentioned here.

Line 124: typo "individual at both at 12"

Line 136: This might need to be explained a little better. It's not clear how the lower 10th percentile excludes the lowest tail of the distribution

Line 168 and 174: I am not clear why endogenous cardiac metabolism was measured last, as my initial instinct would have been to measure it first, as endogenous stores might be utilized even when other substrates are provided, and thus measuring END last might result in an underestime. And why it was measured in the presence of deoxyglucose and iodoacetate? I think something needs to be cleared up in the writing in this section, or perhaps it just needs a little more explanation for the reader who may not be that familiar with these techniques.

Line 197: 10.2g seems pretty large for the average size of a *Fundulus heteroclitus*. Did you intentionally select large/older fish for this experiment?

Line 215: I'm not sure why acclimation order was excluded from the linear models for cardiac metabolism. Although you don't have an estimate of the trait at both acclimation temperatures for each individual, each individual did experience different acclimation orders, so you should be able to compare, for example the cardiac metabolic rate of 12C acclimated fish between fish that were exposed to 12C vs fish that were exposed to 28C first.

Line 243-248: As I mentioned in my major comments, I find the acclimation order effect extremely interesting, and I would be inclined to provide these data in the main text.

Line 251-253: Looking at the raw data, and the data plotted in figure 2 (and figure 3 for that matter), it looks like you have a much smaller dataset to allow you to calculate plasticity (i.e. matched data for the trait at both acclimation temperatures). I think this sample size should be clearly stated somewhere. I think the best place for it would be in the legends of Figures 2 and 3. For CTMax from looking at the raw data file there are matched points for ~90 individuals, so about half of the complete dataset. I think the reader needs to be clearly told this.

Line 254-256: This is exactly the pattern you would expect if there is unaccounted for intra-individual variation, and you are observing regression towards the mean.

Line 270: Should this be Fig S2?

Line 311: I found this text in combination with Figure 5 to be a bit confusing. The figure shows Cardiosomatic index, and you can only derive the conclusion reported in the text (that the hearts were bigger on an absolute basis) by taking into account the parenthetical statement in the figure legend that body mass was the same. The title of the figure is thus also a little unclear. Consider redoing the figure as two regression lines of heart mass against body mass for the two acclimation temperatures and reporting CSI in the text (i.e. using the presentation from Fig S5).

Line 343: I think it would be worth explaining a bit more about the difference between your north reference site and the Dayan et al site. You should make clear the geographic distance between them. If google maps can be relied on, it looks like the Dayan reference site is about 20km further north than your reference site. This is potentially interesting and worthy of comment, given the steep change in allele frequencies and embryonic phenotypic in this part of the species range. Perhaps there is a very steep cline in CTMax in this region, which would be a very cool observation! (and a little different from what was observed by Healy et al (2018) who did not find much difference in CTMax across New Jersey, but did find a steep cline in hypoxia tolerance).

Line 363-365 seems a bit speculative at this point

Line 560-565: I found this analysis and the discussion of the resulting patterns confusing. I think this needs a bit more clarification, as I am having trouble conceptualizing an underlying mechanism that could account for this pattern.

Line 571: This is one of the places where I think that a discussion of the various alternative ways of looking at the data would be helpful. I think a trade-off is only one possible explanation for the patterns.

Line 589: I found the use of the word "obfuscated" interesting. Are you implying that there are actually differences between the populations that you didn't have power to detect? It seems to me that given your pretty robust sample sizes, if there was actually a biologically relevant difference between these populations, you would have detected it.

Review form: Reviewer 2

Is the manuscript scientifically sound in its present form?

Yes

Are the interpretations and conclusions justified by the results?

Yes

Is the language acceptable?

Yes

Do you have any ethical concerns with this paper?

No

Have you any concerns about statistical analyses in this paper?

No

Recommendation?

Major revision is needed (please make suggestions in comments)

Comments to the Author(s)

Congratulations on submitting your manuscript for publication, especially during such an unusual year. There is a wealth of data here and is quite impressive. The main issues I have are that there are areas that need clarification and reorganization. The attachment (see Appendix A) offer suggestions to help in that endeavor. I only recommend major revision to ensure there is enough time to incorporate the changes to your manuscript.

Decision letter (RSOS-210440.R0)

Dear Ms Drown

The Editors assigned to your paper RSOS-210440 "Physiological Plasticity among Populations Subjected to Recent Anthropogenic Heating" have now received comments from reviewers and would like you to revise the paper in accordance with the reviewer comments and any comments from the Editors. Please note this decision does not guarantee eventual acceptance.

Please submit your revised manuscript and required files (see below) no later than 21 days from today's (ie 28-May-2021) date. Note: the ScholarOne system will 'lock' if submission of the revision is attempted 21 or more days after the deadline. If you do not think you will be able to meet this deadline please contact the editorial office immediately.

on behalf of Professor Andrew Simons (Associate Editor) and Kevin Padian (Subject Editor)
openscience@royalsociety.org

Subject Editor Comments to Author (Professor Kevin Padian):

Comments to the Author:

Thanks for your submission. The reviewers are generally positive but have a lot of concerns, so we would like to ask you to address these in a revision and be sure to detail your responses in a separate document. Best wishes.

Associate Editor Comments to Author (Professor Andrew Simons):

Comments to the Author:

Both reviewers are enthusiastic about the perspective, and the high quality and the novelty of the data presented in this manuscript. However, both find that substantial revisions will be required. Especially important is Reviewer 1's compelling argument that acclimation may have been incomplete for the low temperatures, and thus trial order might explain some fraction of observed intraspecific variation. The relevant supplementary table and figure S1 should be included in the main text, and results interpreted in the light of what this reviewer details. Please also discuss the possibility that biological (as opposed to technical) variation is not accounted for because single estimates were obtained per individual per temperature. I also agree that the title should be altered to better reflect findings. Reviewer 2 makes many detailed and worthwhile recommendations to improve clarity and logical flow, and I encourage the authors to consider these suggestions carefully, and to incorporate recommendations so as to improve the quality of the manuscript.

There are a couple of small things that the authors should address in addition to points raised by reviewers. The authors infer that the high interindividual variation observed may be adaptive.

This inference requires further justification and explanation. For example, the authors should point out that to infer that variation is adaptive would require a demonstration that this variation is expressed within, as opposed to only among families/genotypes, since adaptations are not expressed by populations. Also, for a general readership, the abstract should include a couple of words to provide a clue about the species under study beyond the species binomial ("small estuarine fish", for example, is used in Introduction).

Reviewer comments to Author:

Reviewer: 1

Comments to the Author(s)

General Comments

This manuscript reports on an impressive dataset cataloging the extent of inter-individual variation in the acclimation responses in a variety of physiological traits in Atlantic killifish. Although between-population variation in plasticity of physiological traits has been described in a variety of species, inter-individual variation within populations has rarely been characterized, particularly in fish (although see Morgan, Finnogen and Jutfelt 2018), and thus this paper represents a significant advance in our understanding of this important topic.

The particular strengths of the study are the large sample size, the fact that multiple traits were measured for each individual (allowing for correlations among traits to be examined), and the high quality of the methods and statistical approaches used to assess the traits.

I found the paper to be generally clear and easy to follow.

I have only one major comment for the authors to consider. It does not really affect the integrity of the data, but it does suggest some potentially different interpretations. Specifically, I am concerned about the patterns apparent in the data presented in Supplemental Figures 1 and 2, which show that both metabolic rate and CTMax differ depending on the order of acclimation.

For example, CTMax at 12C is higher in individuals that were previously exposed to 28C acclimation. [Note that because of the design, the fish were initially brought through a pseudo winter at 10C, so fish that were acclimated to and tested at 12C first only underwent acclimation from 10 to 12C during the four-week acclimation period, whereas the fish that were first acclimated and tested at 28C and then acclimated to 12C had to undergo acclimation down from 28C to 12C in four weeks.]

In the supplemental results section, the authors suggest that the observed acclimation order effect is due to prior heat hardening because those fish were also exposed to a CTMax trial at the 28C acclimation temperature. However, there is another plausible explanation – that the fish that underwent acclimation to 12C from 28C were not yet fully acclimated to the lower temperature at the time of measurement.

Similar acclimation order effects are evident for metabolic rate, and these are unlikely to be explained by heat hardening due to the prior CTMax trial.

The authors consider and dismiss the possibility of incomplete low temperature acclimation in the supplemental results section based on data presented in Healy and Schulte (2012). However, the data in Healy and Schulte (2012) suggest that at 3 weeks acclimation of CTMax is not yet complete when the direction of acclimation is from high to low temperature and they do not report data on acclimations beyond this time. In general, in fish, acclimation to low temperatures is much slower than is acclimation to higher temperatures, and therefore this is an important consideration. As a result, I would hesitate to entirely dismiss the possibility that a significant fraction of the intraspecific variation in acclimation capacity that has been observed in this study may be the result of trial order. This idea is supported by the fact that trial order is a significant factor in the linear models.

I worry about the extent to which the patterns shown in Figures 2A and Figure 3 are being influenced by this trial order effect. If I look at the spread in the CTMax data, for example, it

seems likely that the majority of the individuals with high CTMax at 12C (and thus low plasticity) are from the group that experienced 28-12C and vice versa. Indeed, the linear models reveal a strong effect of acclimation order on the models for whole animal metabolic rate, CTMax, and Cardiac metabolic rate. It is possible that this result is still robust, but it is difficult to know from the data as presented. It might be interesting to show this in the figures either using different shaped dots or different colours for the different trial orders. This would go a long way towards reassuring the reader of the origins of the observed patterns. Note that it does not change the pattern. But it might alter how you would interpret it.

Another interesting idea that follows from the possibility that low temperature acclimation might not be complete in the fish that were tested first at 28 and then at 12 is that individuals might vary in the rate of acclimation. That is, what if some individuals are not yet fully acclimated, while other have achieved complete acclimation? Thus, the authors might consider whether their observed results are consistent with differences in the capacity for acclimation (as they suggest), or might instead reflect inter-individual differences in the rate of acclimation to low temperature.

Whatever, the case, I think it is absolutely critical that the issue of the effects of acclimation order be brought forward more clearly in the main text and discussed, and their implications for the conclusions drawn be fully considered. Otherwise, a casual reader of the paper who does not delve deeply into the supplements may miss the importance of these data. For example, they do not appear in the abstract.

Finally, a less interesting, less likely, but also possible, explanation for the observed patterns could stem from intra-individual variation in the traits that is not being captured that might influence the assessment of plasticity. Each trait is measured only once on each individual at each acclimation temperature, so the repeatability of the trait is not assessed here. How can you dismiss the possibility that variation in the plastic responses could be due to an inaccurate measure for an individual at one acclimation temperature and then regression/reversion to the mean at the other temperature? For example, imagine that (for whatever reason) the one measure of CTMax was unusually high for an individual acclimated at 12C (i.e. subsequent assessment of the trait in the same individual would have resulted in a higher value for CTMax at 12). If the assessments for this individual were close to the group mean at 28C, then you would get a low value for plasticity that might not actually be real. Although CTMax is thought to be pretty repeatable, the repeatability estimates are only in the range of 0.28-0.65 (Morgan et al 2018). If you modeled the data with this level of "error" in the estimates, how much would that change your relationships? On the other hand, Morgan et al (2018) found similar patterns where individuals with high innate tolerance have smaller acclimation responses, which helps to bolster the argument that this is a real pattern, rather than a reflection of error in the data (although regression towards the mean could be a problem in their data as well). For CTMax, if random variation is greater at 12C than at 28C acclimation, this would also play a role.

I think the potential for random error should be acknowledged a little more in the paper. Technical variation is considered and dismissed in several places, but biological variation isn't really discussed. I agree with the authors that their methods are outstanding and thus the likelihood of technical variation being the cause of the differences are low, but I am less sanguine about dismissing random biological factors given the known variation (moderate level of repeatability) in assessments of traits such as CTMax.

In addition to this major comment above, I have some minor suggestions for revision, outlined below, that occurred to me as a read the paper.

Line 1: I think you should consider altering the title. Although your initial design was intended to examine among-population variation in plasticity, that's not what you found. Instead, you found

high levels of within-population (inter-individual) variation in plasticity. This inter-individual variation in plasticity is really interesting and is an important result, and I think it should be reflected in the title of the paper.

Line 17: The abbreviations for whole animal metabolic rate, critical thermal maximum and cardiac metabolic rate should be provided in brackets after each of the terms. Otherwise, the rest of the abstract is a bit difficult to follow.

Line 19: I know that it is impossible to fully explain your sample sizes, as they varied across traits and acclimation temperatures, but when I look at (for example) CTMax in the raw data, it looks like you only have matched 12 and 28C data for about 86 fish (about half the number reported in the abstract) and it is this matched data that is needed to compute individual plasticity. I haven't checked for the other traits but it looks similar for whole animal metabolic rate. I think a bit more transparency here would be helpful on this issue. I should note that this is still an extremely impressive sample size (even when divided across three populations).

Line 44: You emphasize the daily variation in the habitat of these fish here, but you are looking at plasticity in response to 4-week acclimation to a constant temperature. I think it is also important to also describe the seasonal variation in temperature in these habitats, since the type of plasticity you are looking at is probably more relevant to seasonal acclimatization

Line 66-68: Although I agree that in general interindividual variation in plasticity has received limited attention in animals, it has actually been looked at a fair bit in plants. As a result, I would maybe soften this statement a bit.

See for example:

van Rooijen, R., Aarts, M. G., & Harbinson, J. (2015). Natural genetic variation for acclimation of photosynthetic light use efficiency to growth irradiance in *Arabidopsis*. *Plant Physiology*, 167(4), 1412-1429.

Le, M. Q., Engelsberger, W. R., & Hinch, D. K. (2008). Natural genetic variation in acclimation capacity at sub-zero temperatures after cold acclimation at 4 C in different *Arabidopsis thaliana* accessions. *Cryobiology*, 57(2), 104-112.

And there is the paper by Morgan et al (2018) which you cite elsewhere in the paper that looked at interindividual variation in acclimation of CTMax in fish, which should probably be mentioned here.

Line 124: typo "individual at both at 12"

Line 136: This might need to be explained a little better. It's not clear how the lower 10th percentile excludes the lowest tail of the distribution

Line 168 and 174: I am not clear why endogenous cardiac metabolism was measured last, as my initial instinct would have been to measure it first, as endogenous stores might be utilized even when other substrates are provided, and thus measuring END last might result in an underestimate. And why it was measured in the presence of deoxyglucose and iodoacetate? I think something needs to be cleared up in the writing in this section, or perhaps it just needs a little more explanation for the reader who may not be that familiar with these techniques.

Line 197: 10.2g seems pretty large for the average size of a *Fundulus heteroclitus*. Did you intentionally select large/older fish for this experiment?

Line 215: I'm not sure why acclimation order was excluded from the linear models for cardiac metabolism. Although you don't have an estimate of the trait at both acclimation temperatures for each individual, each individual did experience different acclimation orders, so you should be able to compare, for example the cardiac metabolic rate of 12C acclimated fish between fish that were exposed to 12C vs fish that were exposed to 28C first.

Line 243-248: As I mentioned in my major comments, I find the acclimation order effect extremely interesting, and I would be inclined to provide these data in the main text.

Line 251-253: Looking at the raw data, and the data plotted in figure 2 (and figure 3 for that matter), it looks like you have a much smaller dataset to allow you to calculate plasticity (i.e. matched data for the trait at both acclimation temperatures). I think this sample size should be clearly stated somewhere. I think the best place for it would be in the legends of Figures 2 and 3. For CTMax from looking at the raw data file there are matched points for ~90 individuals, so about half of the complete dataset. I think the reader needs to be clearly told this.

Line 254-256: This is exactly the pattern you would expect if there is unaccounted for intra-individual variation, and you are observing regression towards the mean.

Line 270: Should this be Fig S2?

Line 311: I found this text in combination with Figure 5 to be a bit confusing. The figure shows Cardiosomatic index, and you can only derive the conclusion reported in the text (that the hearts were bigger on an absolute basis) by taking into account the parenthetical statement in the figure legend that body mass was the same. The title of the figure is thus also a little unclear. Consider redoing the figure as two regression lines of heart mass against body mass for the two acclimation temperatures and reporting CSI in the text (i.e. using the presentation from Fig S5).

Line 343: I think it would be worth explaining a bit more about the difference between your north reference site and the Dayan et al site. You should make clear the geographic distance between them. If google maps can be relied on, it looks like the Dayan reference site is about 20km further north than your reference site. This is potentially interesting and worthy of comment, given the steep change in allele frequencies and embryonic phenotypic in this part of the species range. Perhaps there is a very steep cline in CTMax in this region, which would be a very cool observation! (and a little different from what was observed by Healy et al (2018) who did not find much difference in CTMax across New Jersey, but did find a steep cline in hypoxia tolerance).

Line 363-365 seems a bit speculative at this point

Line 560-565: I found this analysis and the discussion of the resulting patterns confusing. I think this needs a bit more clarification, as I am having trouble conceptualizing an underlying mechanism that could account for this pattern.

Line 571: This is one of the places where I think that a discussion of the various alternative ways of looking at the data would be helpful. I think a trade-off is only one possible explanation for the patterns.

Line 589: I found the use of the word "obfuscated" interesting. Are you implying that there are actually differences between the populations that you didn't have power to detect? It seems to me that given your pretty robust sample sizes, if there was actually a biologically relevant difference between these populations, you would have detected it.

Reviewer: 2

Comments to the Author(s)

Congratulations on submitting your manuscript for publication, especially during such an unusual year. There is a wealth of data here and is quite impressive. The main issues I have are that there are areas that need clarification and reorganization. The attachment offer suggestions to help in that endeavor. I only recommend major revision to ensure there is enough time to incorporate the changes to your manuscript.

===PREPARING YOUR MANUSCRIPT===

===PREPARING YOUR REVISION IN SCHOLARONE===

Author's Response to Decision Letter for (RSOS-210440.R0)

See Appendix B.

Decision letter (RSOS-210440.R1)

Dear Ms Drown,

It is a pleasure to accept your manuscript entitled "Interindividual Plasticity in Metabolic and Thermal Tolerance traits from Populations Subjected to Recent Anthropogenic Heating" in its current form for publication in Royal Society Open Science. The comments of the reviewers who reviewed your manuscript are included at the foot of this letter.

on behalf of Professor Andrew Simons (Associate Editor) and Kevin Padian (Subject Editor)
openscience@royalsociety.org

Associate Editor Comments to Author (Professor Andrew Simons):

The authors have adequately addressed concerns raised by reviewers. In particular, a discussion of the effects of acclimation order are now included, as requested by Reviewer 1. The incorporation of these results in existing figures 2 and 3 is a satisfactory alternative to the reviewer's suggestion that Figure S1 be moved to the main text.

Several minor errors were introduced in the revisions, and should be corrected at the proofing stage:

P. 5, l. 150-153: Revise sentence "To capture a ... were used to estimate..." because there are redundant verbs.

l. 271 and l. 298 (should be “temperatures”); l. 376 (should be “differences”); l. 377 (should be “measured”); l. 458 (delete the colon after “due”); l. 460 (please re-word “measurement of traits that are difficult or evolve or not selectively important”); l. 483 (should be “that also show divergence in physiological traits could be compared”); l. 668 & 715 (“yet” should be “however”).

l. 313-314 The sentence “...there was still interindividual variation in the degree of acclimation response...” is not informative as written and should either be deleted or revised. The significance of error variance cannot be tested. Is an argument being made for 9.69% and 13.35% being more than expected?

Appendix A

Firstly, I would like to congratulate you on your dataset, it is rather impressive and it shows that a lot of work went into this project (both in data collection and analysis)! The techniques and data analysis are really interesting and contribute to the know literature. To briefly summarize, your manuscript is looking at the interactions between thermal acclimation and phenotypic plasticity in wild-caught Atlantic killifish. Killifish were sourced from 1) a site known to have higher water temperatures due to anthropogenic heating, 2) a northern reference site, and 3) a southern reference site. Physiological differences did not appear to vary much across population but this was largely attributed to a wide range of inter-individual variation in response to thermal acclimation within the population for a given trait. The inter-individual variation appears to be set by acclimation to colder temperatures (12°C) rather than warmer temperatures. The introduction and discussion suffer from some organizational issues and lack of clarification which I have included in my suggestions below. I only recommend major revision to ensure there is enough time to make the suggested changes. Good luck and I look forward to reading the final manuscript.

* Comments are in order of appearance in the MS, not arranged in order of severity/importance.

** Bolded statements indicate important concerns

Abstract:

Overall the abstract is nicely written and very detailed, however it abruptly ends. Consider adding a final statement about the implications/usefulness of your manuscript (e.g., Studies that focus on population dynamics should account for inter-individual variation that occurs during thermal acclimation because...)

Introduction

Overall the introduction does an adequate job in presenting the current literature, good job! However the main problem with the introduction is that the information that is presented is not organized or polished. Some simple rearranging of the paragraphs and add details to the ideas that have been introduced will help with this. For more details, look at some of the suggestions below.

Line 34 and various places throughout document. Do not start sentence with because, find some other transition word (e.g., yet, since, however, due too...) This makes the thought following the word because appear more polished/complete.

I am not sure how your second paragraph fits into the story of your manuscript. You talk about genetics masking phenotype and how organisms can hide which can result in maladaptive or adaptive responses. But you do not measure any genetic markers or try to ascertain if your populations of animals have maladaptive or adaptive responses. I would suggest removing this paragraph. Then take the third paragraph, divide into sections and expand on the information you have. Try laying your introduction out like this: 1) climate change and anthropogenic heating 2) broad physiological responses 3) acclimation, phenotypic plasticity, and the interactions/drivers 4) specific details about your model organism and study.

Line 68: Make this sentence into two statements. Here we propose a,b,c. We expect to find x,y,z. These statements could be the lead into your final paragraph.

Line 76: Is it CTMax or CT_{max}? I have seen it published both ways, but more frequently and recently as CT_{max}. Both abbreviations mean the same thing. Just be aware that more recent studies are using the CT_{max} abbreviation, possibly to distinguish between CT_{min}

Line 78: None of your listed references (24-26) talk about CaM (what is this abbreviation for? Its unclear) substrates and their relevance as a physiological marker for thermal studies. You need a few lines of justification here to explain what they are and how they relate to cardiac remodeling, which you later talk about in your discussion. In generally animal metabolic rate, CTmax and changes in the concentration of metabolic substrates need to be discussed more since you are using these parameters for the basis of your analysis. This would be good to add to that broad physiological response paragraph (see above). Endogenous... what do you mean by that and how is that relevant?

-Edited to add that this section made sense once I got to the methods, but prior to reading it was really unclear and confusing. Still consider elaborating this section in the introduction. Something along the lines of cardiac tissue metabolic in the presence and absence of aerobic substrates and then reference your methods paper.

Your 3rd paragraph is rather large and would be easier to digest if broken up (see above) and more details/justification was added. Line 89: when you introduce the aims of your project you should also state how you are going to do this. For example we aim to investigate the variation within and among populations by.... And so on with your subsequent aims.

Line 98: Assuming reasonably heritability... Hmm isn't this a big assumption here considering that the heritability of these traits was not studied in killifish so you can't be too sure. Also at least one of the listed references suggest that heritability can be low or largely explained by other factors. Essentially there is not a clear cut answer on this unless directly studied in your model organism. I would remove this statement.

Methods

Line 104: Was this all from the same estuary system? You stated in your intro that estuaries can be highly variable from season to season but also among different estuaries.

Line 109: Common garden is not a term that is typical found in physiology papers, may want to define this for non-genetic readers.

Line 133: Consider moving calculations/equations to the statistics section rather than embedded within the experimental methods.

Line 146: Probably should justify the rate of temp change, that it was what was used in previously published studies and has been shown to prevent lag between body and water temperature. Becker, C.

D. & Genoway, R. G. Evaluation of the critical thermal maximum for determining thermal tolerance of freshwater fish. *Environmental Biology of Fishes* 4, 245–256 (1979).

Calculations and statistical analysis seem sound and follow standard methods.

Results

Line 224: You did not talk about how water parameters were collected when you collected fish, so this paragraph comes out of left-field. Consider adding a brief section in your methods that discusses this.

Line 243: Define what you mean by acclimation order in the beginning of the paragraph rather than the end. Reader is left wondering what is acclimation order and how did they miss it and did the author mention it before.

Line 253: variation patterns are acclimation at 12°C and a low response will have a high response at 28°C and vice versa... did these responses correlate with catch site?

Line 301 states CaM was unaffected by acclimation temperature but line 322 states that CaM was one of the more variable traits? How do you reconcile this, these statements seem mutually exclusive? Is it because less variation was explained by thermal acclimation in the CaM method compared to the others? If this is so, this needs to be clarified.

Discussion

Summary of the first paragraph: despite prior evidence for divergence, there was no evidence in this population and the authors are unsure about the cause. It could be related to migration (although less so), thermal difference is not great enough to illicit change (although less so), traits are difficult to evolve, not selectively important, or trait variation is important for survival. The way to figure out which of these it is, is by doing a genetic analysis.

-Would generation and epigenetics play a role in trait divergence and lack of divergence? Might want to consider these articles and incorporating them into your first paragraph. 1) Burggren, W. (2016). Epigenetic inheritance and its role in evolutionary biology: re-evaluation and new perspectives. *Biology*, 5(2), 24. 2) Ho, D. H., & Burggren, W. W. (2010). Epigenetics and transgenerational transfer: a physiological perspective. *Journal of Experimental Biology*, 213(1), 3-16.

-This paragraph could be organized to flow a little bit better. After talking about divergence and lack of divergence consider, talking about why that is (the five suggestion layed out above), and then go into details about the likelihood of each possible.

Second paragraph of discussion: Consider add a intro sentence at the beginning of the first paragraph under each header. Remind the reader what is inter-individual variation and why it is significant then talk about what you found. The first half of this section is a retelling of the results section, which the reader can skip and beginning at the second paragraph and not miss much since it is a retelling.

Line 388: when you say reported by other teleost fish...immediately follow that with all the reference you are about to get into detail with. Reported by other teleost fish (reference number to reference number). You need to support that idea that is has been previously reported before getting into the details.

Line 392: why is greater individual variation important for evolutionary history? I rather read about more about that, then validation of the technique which happens in the subsequent sentences. Considering making a separate header section (1st paragraph of discussion) that goes into the validation of this technique and why your results are due to biological variation, get that out of the way first. Then the following paragraphs can be spent telling the reader why this is exciting and what it means from an evolutionary perspective.

Line 470:....what is your evidence for cardiac remodeling. This is the first I recall hearing about it. You need to reference back to your data and build a case for it or use previously published studies that support this. Hey I found the evidence that supports this at Line 508! So you should probably introduce that sooner.

Paragraph at 505: I think I recall reading that you were not able to collect mass for all hearts. Which is a real shame. That would mean that your CaM is not normalized/standardized to tissue mass then? What is CaM standardized too? I only ask because if that is the case you would definitely know the answer to the statement in line 510.

Line 533: Hmm while some species have a limited capacity to alter metabolic rate others do not. 1) KIRBY, A. R., CROSSLEY, D. A. & MAGER, E. M. 2020. The metabolism and swimming performance of sheepshead minnows (*Cyprinodon variegatus*) following thermal acclimation or acute thermal exposure. *Journal of Comparative Physiology B*. 2) PILAKOUTA, N., KILLEN, S. S., KRISTJÁNSSON, B. K., SKÚLASON, S., LINDSTRÖM, J., METCALFE, N. B. & PARSONS, K. J. 2020. Multigenerational exposure to elevated temperatures leads to a reduction in standard metabolic rate in the wild. *Functional Ecology*, 34, 1205-1214. 3) SANDBLOM, E., GRÄNS, A., AXELSSON, M. & SETH, H. 2014. Temperature acclimation rate of aerobic scope and feeding metabolism in fishes: implications in a thermally extreme future. *Proceedings of the Royal Society of London B: Biological Sciences*, 281, 20141490. 4) MCDONNELL, L. H. & CHAPMAN, L. J. 2016. Effects of thermal increase on aerobic capacity and swim performance in a tropical inland fish. *Comparative Biochemistry and Physiology Part A: Molecular & Integrative Physiology*, 199, 62-70. 5) RUMMER, J. L., COUTURIER, C. S., STECYK, J. A. W., GARDINER, N. M., KINCH, J. P., NILSSON, G. E. & MUNDAY, P. L. 2014. Life on the edge: thermal optima for aerobic scope of equatorial reef fishes are close to current day temperatures. *Global Change Biology*, 20, 1055-1066.

Paragraph at 539: This is really novel and should be discussed/highlighted more. Particularly mention this in the abstract instead of buried in the discussion. And your novel findings want to be the bulk of your discussion not validation of experimental methods. This paper about the sheepshead minnow an ecologically similar species may be helpful here. FANGUE, N. A., WUNDERLY, M. A., DABRUZZI, T. F. & BENNETT, W. A. 2014. Asymmetric Thermal Acclimation Responses Allow Sheepshead Minnow *Cyprinodon variegatus* to Cope with Rapidly Changing Temperatures. *Physiological and Biochemical Zoology*, 87, 805-816.

Line 587: Conclusion needs a stronger/global impact and ending about the significance of the data within the realm of the published literature. Why the findings are novel.

Appendix B

Dear Dr. Simons,

Thank you for considering our submitted manuscript, now entitled “*Inter-individual Plasticity in Metabolic and Thermal Tolerance traits from Populations Subjected to Recent Anthropogenic Heating*”. We have made revisions and thank both reviewers for their comments, which have improved the manuscript. Responses to all editor and reviewer comments have been made (detailed below in blue), and a marked manuscript that shows changes in blue has been provided. Specific comments also contain reference to specific relevant lines in the marked manuscript.

The most significant changes were made to address one major comment made by reviewer one with regard to the acclimation order effect on physiological plasticity in whole animal metabolic rate and critical thermal maximum. **This major comment has been addressed in four ways:**

- 1) These results are now mentioned in the abstract (lines 23-25).
- 2) As suggested by the reviewer, Figures 2A, 2B, 2D, and 3 now include the response for each acclimation group including acclimation order (circles and triangles). Figure legends have also been edited to include the effect of acclimation order on plasticity in xCT_{max} and WAM.
- 3) Linear models, as described for the other physiological traits, have been used to analyze the effects of sex, body mass, acclimation order, and all second order interaction terms on plasticity in CT_{max} and WAM. The results of these linear models have been added on Lines 286-291 (WAM) and 311-315 (CT_{max}) and in Table S1. In summary, acclimation order did have a significant effect on plasticity in both CT_{max} and WAM, which informs the interpretation of the large inter-individual variation in plasticity for these traits.
- 4) The interpretation of the large inter-individual variation in plasticity for CT_{max} and WAM has been discussed in light of the results described above. Part of this discussion was previously included in the supplemental results section and has been moved to the main text discussion (now beginning on line 535). This section has been revised to reflect the results described above and addresses two possible explanations for these data. This discussion is found primarily on lines 556-583 of the marked manuscript. Small revisions have also been made on lines 675, 680, and 682 to incorporate these results into the interpretation of results.

In addition, minor comments from both reviewers have been incorporated. **Most significantly, we expanded and reorganized the discussion section to focus on the novel findings and incorporate additional ways of interpreting the data.** All changes are detailed in response to specific reviewer comments below.

We look forward to future correspondence and thank you for your time and consideration of our manuscript.

Best wishes,

Melissa Drown

mxdl288@miami.edu

Editor Comments to the Author:

Both reviewers are enthusiastic about the perspective, and the high quality and the novelty of the data presented in this manuscript. However, both find that substantial revisions will be required. Especially important is Reviewer 1's compelling argument that acclimation may have been incomplete for the low temperatures, and thus trial order might explain some fraction of observed intraspecific variation. The relevant supplementary table and figure S1 should be included in the main text, and results interpreted in the light of what this reviewer details.

We have addressed responses to both reviewers as summarized in our letter and detailed below. Special attention was given to incomplete acclimation at low temperature and biological variation. To summarize, we have modified figures 2 and 3 to show the acclimation order effect and added linear models to analyze the effect of acclimation order (and other co-variables) on plasticity in WAM and CT_{max} . The results of these models are found in supplemental Table 1, in the results on lines 287-292 (WAM) and 312-316 (CT_{max}), and in the discussion on lines 547-573. We have modified the existing main text figures (2 and 3) and included the relevant information in the main text of the results rather than moving Figure S1 and supplemental table 1 to the main text.

Please also discuss the possibility that biological (as opposed to technical) variation is not accounted for because single estimates were obtained per individual per temperature.

Biological variation in acclimation response that could affect our interpretation of the relationships between CT_{max} and WAM with plasticity has been enhanced as detailed in response to reviewer 1 (Major Concern B).

I also agree that the title should be altered to better reflect findings.

We have altered the title; it is now: *“Inter-individual Plasticity in Metabolic and Thermal Tolerance traits from Populations Subjected to Recent Anthropogenic Heating”*

Reviewer 2 makes many detailed and worthwhile recommendations to improve clarity and logical flow, and I encourage the authors to consider these suggestions carefully, and to incorporate recommendations so as to improve the quality of the manuscript.

Several comments from reviewer 2 have been incorporated and have improved the quality of the manuscript. Most significant was the expansion and reorganization of the discussion section. This section now begins with a discussion of technical and biological variation and then proceeds to discuss and interpret the interesting and novel results. Comments made by reviewer 2 contributed most notably to discussion lines 479-488, 649-657, and 754-759 of the marked manuscript. As suggested by reviewer 2, we have also added a short paragraph at the start of the discussion to highlight interesting and novel results (lines 372-378).

There are a couple of small things that the authors should address in addition to points raised by reviewers. The authors infer that the high interindividual variation observed may be adaptive. This inference requires further justification and explanation. For example, the authors should

point out that to infer that variation is adaptive would require a demonstration that this variation is expressed within, as opposed to only among families/genotypes, since adaptations are not expressed by populations.

Thank you for the recommendation; this is addressed on lines 483-485: *“To address the unexpected lack of divergence among populations, genes associated with the high interindividual variance could be identified and patterns in polymorphisms in these genes partitioned among individuals.”*

Also, for a general readership, the abstract should include a couple of words to provide a clue about the species under study beyond the species binomial (“small estuarine fish”, for example, is used in Introduction).

Thank you for the suggestion; we have added “small estuarine fish” after the species binomial in the abstract on line 12.

Additional comment to the editor:

Both reviewers suggested changes to the abstract that we would like to incorporate. In order to do this, our abstract has been lengthened in the marked and unmarked manuscript. This included the addition of a brief description of the species (small estuarine fish) on line 12, addition of a final sentence stating the research importance (lines 34-36), and highlighting the novel findings of the manuscript more clearly (lines 22-26). After incorporating these changes, the abstract is 346 words, which exceeds the abstract word limit. We would like to request an exception to the abstract word limit in this case to accommodate these changes, which greatly improve the quality of the abstract and will make the broad importance of our research more clear to potential readers.

REVIEWER COMMENTS TO AUTHOR:

REVIEWER: 1

General Comments *(from Reviewer 1):*

This manuscript reports on an impressive dataset cataloging the extent of inter-individual variation in the acclimation responses in a variety of physiological traits in Atlantic killifish. Although between-population variation in plasticity of physiological traits has been described in a variety of species, inter-individual variation within populations has rarely been characterized, particularly in fish (although see Morgan, Finnogen and Jutfelt 2018), and thus this paper represents a significant advance in our understanding of this important topic.

The particular strengths of the study are the large sample size, the fact that multiple traits were measured for each individual (allowing for correlations among traits to be examined), and the high quality of the methods and statistical approaches used to assess the traits.

I found the paper to be generally clear and easy to follow.

I have only one major comment for the authors to consider. It does not really affect the integrity of the data, but it does suggest some potentially different interpretations. Specifically, I am concerned about the patterns apparent in the data presented in Supplemental Figures 1 and 2, which show that both metabolic rate and CTMax differ depending on the order of acclimation.

For example, CTMax at 12C is higher in individuals that were previously exposed to 28C acclimation. [Note that because of the design, the fish were initially brought through a pseudo winter at 10C, so fish that were acclimated to and tested at 12C first only underwent acclimation from 10 to 12C during the four-week acclimation period, whereas the fish that were first acclimated and tested at 28C and then acclimated to 12C had to undergo acclimation down from 28C to 12C in four weeks.]

In the supplemental results section, the authors suggest that the observed acclimation order effect is due to prior heat hardening because those fish were also exposed to a CTMax trial at the 28C acclimation temperature. However, there is another plausible explanation – that the fish that underwent acclimation to 12C from 28C were not yet fully acclimated to the lower temperature at the time of measurement.

Similar acclimation order effects are evident for metabolic rate, and these are unlikely to be explained by heat hardening due to the prior CTMax trial.

The authors consider and dismiss the possibility of incomplete low temperature acclimation in the supplemental results section based on data presented in Healy and Schulte (2012). However, the data in Healy and Schulte (2012) suggest that at 3 weeks acclimation of CTMax is not yet complete when the direction of acclimation is from high to low temperature, and they do not report data on acclimations beyond this time. In general, in fish, acclimation to low temperatures is much slower than is acclimation to higher temperatures, and therefore this is an important consideration. As a result, I would hesitate to entirely dismiss the possibility that a significant fraction of the intraspecific variation in acclimation capacity that has been observed in this study may be the result of trial order. This idea is supported by the fact that trial order is a significant factor in the linear models.

I worry about the extent to which the patterns shown in Figures 2A and Figure 3 are being influenced by this trial order effect. If I look at the spread in the CTMax data, for example, it seems likely that the majority of the individuals with high CTMax at 12C (and thus low plasticity) are from the group that experienced 28-12C and vice versa. Indeed, the linear models reveal a strong effect of acclimation order on the models for whole animal metabolic rate, CTMax, and Cardiac metabolic rate. It is possible that this result is still robust, but it is difficult to know from the data as presented. It might be interesting to show this in the figures either using different shaped dots or different colours for the different trial orders. This would go a long way towards reassuring the reader of the origins of the observed patterns. Note that it does not change the pattern. But it might alter how you would interpret it.

Another interesting idea that follows from the possibility that low temperature acclimation might not be complete in the fish that were tested first at 28 and then at 12 is that individuals might

vary in the rate of acclimation. That is, what if some individuals are not yet fully acclimated, while others have achieved complete acclimation? Thus, the authors might consider whether their observed results are consistent with differences in the capacity for acclimation (as they suggest), or might instead reflect inter-individual differences in the rate of acclimation to low temperature.

Whatever the case, I think it is absolutely critical that the issue of the effects of acclimation order be brought forward more clearly in the main text and discussed, and their implications for the conclusions drawn be fully considered. Otherwise, a casual reader of the paper who does not delve deeply into the supplements may miss the importance of these data. For example, they do not appear in the abstract.

Finally, a less interesting, less likely, but also possible, explanation for the observed patterns could stem from intra-individual variation in the traits that is not being captured that might influence the assessment of plasticity. Each trait is measured only once on each individual at each acclimation temperature, so the repeatability of the trait is not assessed here. How can you dismiss the possibility that variation in the plastic responses could be due to an inaccurate measure for an individual at one acclimation temperature and then regression/reversion to the mean at the other temperature? For example, imagine that (for whatever reason) the one measure of CTMax was unusually high for an individual acclimated at 12C (i.e. subsequent assessment of the trait in the same individual would have resulted in a higher value for CTMax at 12). If the assessments for this individual were close to the group mean at 28C, then you would get a low value for plasticity that might not actually be real. Although CTMax is thought to be pretty repeatable, the repeatability estimates are only in the range of 0.28-0.65 (Morgan et al 2018). If you modeled the data with this level of “error” in the estimates, how much would that change your relationships? On the other hand, Morgan et al (2018) found similar patterns where individuals with high innate tolerance have smaller acclimation responses, which helps to bolster the argument that this is a real pattern, rather than a reflection of error in the data (although regression towards the mean could be a problem in their data as well). For CTMax, if random variation is greater at 12C than at 28C acclimation, this would also play a role.

I think the potential for random error should be acknowledged a little more in the paper. Technical variation is considered and dismissed in several places, but biological variation isn't really discussed. I agree with the authors that their methods are outstanding and thus the likelihood of technical variation being the cause of the differences are low, but I am less sanguine about dismissing random biological factors given the known variation (moderate level of repeatability) in assessments of traits such as CTMax.

We thank the reviewer for their thoughtful consideration and comments which have improved the interpretation of the data and the overall manuscript. We divide these comments from Reviewer 1 to address A) acclimation order and the effectiveness and time to complete acclimation first. Then address B) The effect of intra-individual (within individual) variation as it relates to plasticity.

A: Acclimation Order

Reviewer 1: *I have only one major comment for the authors to consider. It does not really affect the integrity of the data, but it does suggest some potentially different*

interpretations. Specifically, I am concerned about the patterns apparent in the data presented in Supplemental Figures 1 and 2, which show that both metabolic rate and CT_{max} differ depending on the order of acclimation.

This major comment has been addressed in four ways:

- 1) These results are now mentioned in the abstract (lines 23-25).
- 2) As suggested by the reviewer, Figures 2A, 2B, 2D and 3 now include different shapes for each acclimation group and figure legends have been edited to include the effect of acclimation order on plasticity in CT_{max} and WAM.
- 3) Linear models, as described for the other physiological traits, have been used to analyze the effects of sex, body mass, acclimation order, and all second order interaction terms on plasticity in CT_{max} and WAM. The results of these linear models have been added on Lines 286-291 (WAM) and 311-315 (CT_{max}) and in Table S1. In summary, acclimation order did have a significant effect on plasticity in both CT_{max} and WAM, which informs the interpretation of the large inter-individual variation in plasticity for these traits. Relevant text is copied below:

Lines 286-291: “Acclimation order also had a significant effect on plasticity in WAM and explained 8.12% of variation among individuals leaving a substantial proportion of variation among individuals unexplained by the variables measured here ($p < 0.05$). However, the observed groupings of individuals based on variation in plasticity (three groups with high, intermediate, or low acclimation response) do not appear to be driven by acclimation order. That is, not all individuals with high, low, or intermediate plasticity were from one acclimation order group.”

Lines 311-315: “Due to the significant effect of acclimation order on 12°C CT_{max}, acclimation order also explained a large and significant proportion of variation in CT_{max} plasticity (38.58%, $p < 0.0001$). Interestingly, within a single acclimation order group, there was still inter-individual variation in the degree of acclimation response with a CV of 9.69% for acclimation order group 1 and 13.35% for acclimation order group 2.”

- 4) The interpretation of the large inter-individual variation in plasticity for CT_{max} and WAM has been discussed in light of the results described above. Part of this discussion was previously included in the supplemental results section and has been moved to the main text discussion (now beginning on line 535). This section has been revised to reflect the results described above and addresses two possible explanations for these data. This discussion is found primarily on lines 556-583 (see text below) of the marked manuscript. Small revisions have also been made on lines 675, 680, and 682 to incorporate these results into the interpretation of results.

*Lines 556-583: “We suggest that while typical acclimation responses may be relatively quick in *F. heteroclitus*, exposure to high, nearly lethal temperatures can have a lasting effect when acclimated to a high temperature. Thus, it may be that the combination of high acclimation temperature and the higher CT_{max} temperature (average CT_{max} 42.5°C at 28°C acclimation) has a lasting effect that is not seen at low acclimation temperature (average CT_{max} 36.3°C at 12°C acclimation). Therefore, higher absolute temperature exposure has a long-lasting residual effect that is not seen when acclimated to low temperature. Interestingly, these same individuals acclimated to 28°C first had a significantly lower WAM at 12°C than individuals acclimated to*

12°C first. Thus, for both CT_{max} and WAM, 12°C performance was affected by previous thermal conditions while 28°C performance was not. This suggests that acclimation to a higher temperature is more effective because performance at the higher acclimation temperature was independent of prior thermal experience.

A corollary to acclimation to higher temperature being more effective is that the 4-week acclimation to 12°C from 28°C was insufficient. Prior studies in *F. heteroclitus* found that acclimation from 15°C to 5°C or 15°C to 25°C occurred within 3 weeks and that acclimation to cooler temperatures occurred more slowly (73). This would explain the significant effect of acclimation order on plasticity in CT_{max} (Fig. S2), which results from a higher 12°C CT_{max} if acclimated to 28°C first (acclimated to 28°C before acclimation to 12°C). Yet, within a single acclimation order group there was still substantial variation in the degree of acclimation response (CV=9.69% for 12 to 28°C, CV=13.35% for 28 to 12°C), and importantly, plasticity is still a function of the 12°C response within each acclimation order group ($R^2 = 0.89$ for 12 to 28°C, $R^2 = 0.87$ for 28 to 12°C, $p < 0.0001$ for both). The similar CV for plasticity in CT_{max} between acclimation groups, as well as the similar significant slope between 12°C CT_{max} versus plasticity, suggests that this variation exists both when acclimating from both warm to cool and cool to warm temperatures. While it is unlikely that the high and nearly lethal temperatures experienced during CT_{max} would occur, the acclimation temperatures used here are ecologically relevant, and thus variation in acclimation response within an acclimation order group represents biologically relevant inter-individual variation.”

B: Intra-individual variation

Reviewer 1: Biological (as opposed to technical) variation that does not account for intra-individual variation is affecting plasticity.

Conceptually, there is within an individual variation that affects how we interpret our data. This would not alter the difference in means among populations or between acclimation groups because it is one of the sources of random errors accounted for in our models. We (the authors) believe that Reviewer 1’s concerns about the effect of intra-individual variation relate to plasticity. Specifically, the statements from the last two paragraphs of Reviewer 1’s “Major concerns”:

Reviewer 1: Finally, a less interesting, less likely, but also possible, explanation for the observed patterns could stem from intra-individual variation in the traits that is not being captured that might influence the assessment of **plasticity**.....

I think the potential for random error should be acknowledged a little more in the paper.

Technical variation is considered and dismissed in several places, but biological variation isn’t really discussed.....

.. For example, imagine that (for whatever reason) the one measure of CT_{max} was unusually high for an individual acclimated at 12C (i.e. subsequent assessment of the trait in the same individual would have resulted in a higher value for CT_{max} at 12). If the assessments for this individual were close to the group mean at 28C, then you would get a low value for **plasticity** that might not actually be real.

And from minor comments in reference to the discussion of WAM.

Reviewer 1: Line 254-256: This is exactly the pattern you would expect if there is unaccounted for intra-individual variation, and you are observing regression towards the mean.

Thus, we believe Review 1's concern about intra-individual variation and plasticity is mostly related to WAM. Although Reviewer 1 clearly provides an example for CT_{max} , the analogy is most closely related to Figure 2C and D, where 6 individuals had low 12°C WAM with average or large 28°C WAM (red Fig. 2C) or low 28°C WAM and average or high 12°C WAM (green Fig 2C).

However, we addressed concerns about the impact of intra-individual (within an individual) variation for the interpretation of CT_{max} (lines 664-673) and WAM (lines 684-724). Relevant lines from the marked manuscript are copied below.

For CT_{max} lines 664-673.

Lines 664-673: "The observation that 93% of the variation in CT_{max} plasticity is explained by 12°C CT_{max} indicates that there is little unaccounted for within individual variation. However, the lack of significant R^2 for 28°C CT_{max} could arise from a large random error or unaccounted within individual variation (e.g., time of day, or unaccounted for stress factors). Yet, this is unlikely because there is little variation in 28°C CT_{max} (Tables 1 and 3) and these 28°C CT_{max} measures exist in plasticity (\log_2 ratio 28/12), which is accounted for by 12°C CT_{max} (Fig. 3). Thus, we suggest that the reason plasticity is unrelated to 28°C CT_{max} is not due to random error or unaccounted within individual variation but instead reflects small variation when approaching the upper-limit of thermal tolerance at higher temperatures."

For WAM, we provide additional comments on the effect of intra-individual variation and then provide an alternative interpretation concerning unaccounted source of variation. We include the complete paragraph (with changes underlined in blue) for clarity.

Lines 684-724: "Finally, there is a similar pattern of variation for acclimation response among individuals for WAM and CT_{max} where low 12°C performance is associated with greater acclimation response (Fig. 2). However, unlike for CT_{max} , plasticity in the WAM acclimation response was a function of both 12°C and 28°C WAM (Fig. 2A and B). Thus, individuals with low 12°C WAM and high 28°C WAM have greater plasticity than individuals with high 12°C WAM and low 28°C WAM. This response, where plasticity is a function of both 12°C WAM and 28°C WAM, is for all 58 individuals with WAM measured at both acclimation temperatures and is similar for all three plasticity groups (Fig. 2C). Unaccounted within individual variation would affect the R^2 (part of the error term of the linear model; therefore, repeated measures of individuals could make the relationship between WAM and plasticity more precise.

While there was a significant relationship between plasticity and both 12°C and 28°C WAM, surprisingly, there was no overall significant correlation between performance at 12°C and 28°C for WAM until individuals were grouped by the magnitude of acclimation responses (lower 10%, middle, and upper 10% confidence interval, or CI, Fig. 2C). Interestingly, the strength of the relationship between performance at 12°C and 28°C also differed, with a significant correlation only in the middle group (between lower and upper 10% CI, $R^2=0.30$); little correlation occurred for the lower 10% CI ($R^2 = 0.04$), and a negative correlation occurred for the upper 10% CI ($R^2 = 0.49$). The significant relationship between 12°C WAM and 28°C WAM is only found among individuals with moderate plasticity and not among

individuals with extreme high or low plasticity. This could reflect differences acclimation response: individuals with very low plasticity (green Fig. 2C and D) tended to have low 28°C WAM, and individuals with very high plasticity (red Fig. 2C & D) tended to have low 12°C WAM. Thus, the potential difference among individuals in acclimation response affects the relationship between 12°C and 28°C WAM.

Alternatively, unaccounted within individual variation may explain why there is no significant relationship between 12°C WAM and 28°C WAM in individuals with extreme plasticity. Specifically, measures of WAM at one temperature for individuals with very low or very high plasticity (upper and lower 10% CI) were impacted by some individual biological effect (e.g., stress response). Thus, for these individuals with extreme plasticity there may have been a lack of repeatability in WAM measurement due to biological and not technical variation impacting the ability to assess acclimation response and correlation between performance at 12°C and 28°C. Yet, these data are similar to CT_{max} and suggest that variation in WAM performance at 12°C affects variation among individuals in acclimation response. These results, where acclimation response is a function of metabolic rate are consistent with a prior study using a coral reef fish, *Lates calcarifer*, where individuals with an initially high metabolic rate acclimation to cooler condition had a smaller acclimation response than individuals with an initially low metabolic rate (99). As previously discussed, this may be due to variation among individuals in the ability to acclimate, especially when going from 28°C to 12°C acclimation conditions. Alternatively, a trade-off between temperature specific responses and the magnitude of acclimation responses could explain the correlation between low temperature performance and magnitude of acclimation response.”

In addition to this major comment above, I have some minor suggestions for revision, outlined below, that occurred to me as a read the paper.

Line 1: I think you should consider altering the title. Although your initial design was intended to examine among-population variation in plasticity, that's not what you found. Instead, you found high levels of within-population (inter-individual) variation in plasticity. This inter-individual variation in plasticity is really interesting and is an important result, and I think it should be reflected in the title of the paper.

The title has been changed to: *“Inter-individual Plasticity in Metabolic and Thermal Tolerance traits from Populations Subjected to Recent Anthropogenic Heating”*

Line 17: The abbreviations for whole animal metabolic rate, critical thermal maximum and cardiac metabolic rate should be provided in brackets after each of the terms. Otherwise, the rest of the abstract is a bit difficult to follow.

Abbreviations have been added to the abstract for clarification (lines 18 and 19).

Line 19: I know that it is impossible to fully explain your sample sizes, as they varied across traits and acclimation temperatures, but when I look at (for example) CT_{max} in the raw data, it looks like you only have matched 12 and 28C data for about 86 fish (about half the number reported in the abstract) and it is this matched data that is needed to compute individual plasticity. I haven't checked for the other traits but it looks similar for whole animal metabolic

rate. I think a bit more transparency here would be helpful on this issue. I should note that this is still an extremely impressive sample size (even when divided across three populations).

It is nearly impossible to describe sample sizes for all comparisons within the word limits for an abstract. While the statement is accurate that ~160 were measured for the traits specified, it is also accurate that comparisons across acclimation temperatures did not include all of ~160 fish. Our suggestion to address this concern is to describe sample sizes for WAM and CT_{max} plasticity on lines 271, 298, figure legends (2, 3 and 4) and in Table S1.

Line 271: "... there was variation among individuals in the degree of WAM acclimation response (data at both acclimation temperature for WAM, N=58)."

Line 298: "... and acclimation order (group) had a significant effect (data at both acclimation temperature for CT_{max} , N=88)."

Line 44: You emphasize the daily variation in the habitat of these fish here, but you are looking at plasticity in response to 4-week acclimation to a constant temperature. I think it is also important to also describe the seasonal variation in temperature in these habitats, since the type of plasticity you are looking at is probably more relevant to seasonal acclimatization

This has been addressed on Lines 46-49: "Seasonally, populations in the northern part of the range may additionally experience temperatures that vary by $>20^{\circ}C$ from summer to winter with some populations likely experiencing freeze-thaw cycles during the winter months."

Line 66-68: Although I agree that in general interindividual variation in plasticity has received limited attention in animals, it has actually been looked at a fair bit in plants. As a result, I would maybe soften this statement a bit.

See for example:

van Rooijen, R., Aarts, M. G., & Harbinson, J. (2015). Natural genetic variation for acclimation of photosynthetic light use efficiency to growth irradiance in Arabidopsis. *Plant Physiology*, 167(4), 1412-1429.

Le, M. Q., Engelsberger, W. R., & Hinch, D. K. (2008). Natural genetic variation in acclimation capacity at sub-zero temperatures after cold acclimation at 4 C in different Arabidopsis thaliana accessions. *Cryobiology*, 57(2), 104-112.

And there is the paper by Morgan et al (2018) which you cite elsewhere in the paper that looked at interindividual variation in acclimation of CT_{Max} in fish, which should probably be mentioned here.

Thank you for the suggested references. To point the reader to relevant literature that may address inter-individual variation in plasticity the suggested references Morgan et al 2018 and some of the plant literature has been cited. In addition, now lines 73-76 have been edited.

Lines 73-76: “Yet, for animals there is little information on the interindividual variation in acclimation responses, how, or whether this variation among traits is related, and the potential evolutionary importance (although see Morgan et al 2018 and plant literature e.g. Le et al 2008, van Rooijen et al 2015, Pazzaglia et al 2021).”

Line 124: typo “individual at both at 12”

Thank you, the typo has been fixed.

Line 136: This might need to be explained a little better. It’s not clear how the lower 10th percentile excludes the lowest tail of the distribution

To clarify, this line has been edited and a reference added. Now lines 150-157:
“To capture a minimum or resting metabolic rate a single value was defined by the 10th percentile values from the cumulative frequency distribution (CFD) of all replicate metabolic rates from each individual (minimum 20 replicate rates of oxygen consumption per time) were used to estimate individual standard metabolic rates (SMR). This 10th percentile value captures the time period when the fish were most at rest during measurement and excludes the lowest tail of the data distribution by selecting a value for SMR that lies on the CFD curve rather than averaging the lowest 10% of data points, which may be sensitive to outliers (42).”

Line 168 and 174: I am not clear why endogenous cardiac metabolism was measured last, as my initial instinct would have been to measure it first, as endogenous stores might be utilized even when other substrates are provided, and thus measuring END last might result in an underestimate. And why it was measured in the presence of deoxyglucose and iodoacetate? I think something needs to be cleared up in the writing in this section, or perhaps it just needs a little more explanation for the reader who may not be that familiar with these techniques.

Yes, other endogenous substrates may have contributed to the measurement of cardiac metabolic rate in the presence of glucose (e.g. glycogen) or FA/LKA (e.g. another non-glycolytic substrate). However, there are two reasons for measuring END last. 1) CaM irreversibly decreases without substrates after 20-30 minutes (unpublished), and this can be avoided by providing exogenous substrates. 2) The intention in measuring END last was to capture non-glycolytic oxygen consumption of the heart (stated on Line 187), which required deoxyglucose and iodoacetate glycolytic enzyme inhibitors (also used for FA and LKA measurements). To clarify this to the reader, more detail has been added earlier in the manuscript (prior to methods description) on lines 84-89. A reference to the methods manuscript was also added on line 86.

Line 86: *For CaM, we measured heart tissue oxygen consumption in the presence of four aerobic substrates: glucose, fatty acids, lactate plus ketones plus ethanol, and endogenous [i.e., non-glycolytic metabolism with no added substrates] (29-31).*

Line 197: 10.2g seems pretty large for the average size of a *Fundulus heteroclitus*. Did you intentionally select large/older fish for this experiment?

Fish captured in September of 2018 for these experiments have been roughly aged using otoliths and were 1-2 years old at the time of the experiment. These individuals were kept in the lab for ~3 months prior to the start of the experiment, during which time they were fed ad libitum, which allowed them to grow substantially. The large size of these fish is reflective of their growth in the lab and not of their size at original collection.

Line 215: I'm not sure why acclimation order was excluded from the linear models for cardiac metabolism. Although you don't have an estimate of the trait at both acclimation temperatures for each individual, each individual did experience different acclimation orders, so you should be able to compare, for example the cardiac metabolic rate of 12C acclimated fish between fish that were exposed to 12C vs fish that were exposed to 28C first.

This has been clarified in the methods on Lines 233-235 of the marked manuscript. Basically, for CaM all individuals measured at 12°C, were initially acclimated to 28°C then acclimated and measured at 12°C and all individuals measured at 28°C were initially acclimated to 12°C then acclimated and measured at 28°C.

Line 233-235: *“For cardiac metabolic rates, substrate was also included as a covariate, but acclimation order was excluded because all individuals assayed at a single acclimation temperature for CaM were from the same acclimation order group.”*

Line 243-248: As I mentioned in my major comments, I find the acclimation order effect extremely interesting, and I would be inclined to provide these data in the main text.

To highlight these data more clearly, Figures 2A, 2B, and 3 now show acclimation order with different shapes. As stated in response to the major comment regarding acclimation order effect, several changes including additional linear models, modification of figures, and addition of a new section to the main text discussion have been made.

Line 251-253: Looking at the raw data, and the data plotted in figure 2 (and figure 3 for that matter), it looks like you have a much smaller dataset to allow you to calculate plasticity (i.e. matched data for the trait at both acclimation temperatures). I think this sample size should be clearly stated somewhere. I think the best place for it would be in the legends of Figures 2 and 3. For CTMax from looking at the raw data file there are matched points for ~90 individuals, so about half of the complete dataset. I think the reader needs to be clearly told this.

We agree, because the sample sizes for plasticity were reduced compared to the complete dataset these sample sizes need to be more clearly defined. We now include these sample sizes the legends of Figure 2 and 3 and in Table S1. In addition, samples sizes have been explicitly stated in the results on Line 271 for WAM and Line 298 for CaM.

Line 254-256: This is exactly the pattern you would expect if there is unaccounted for intra-individual variation, and you are observing regression towards the mean.

We address this above because this comment is also related to the last two paragraph of Reviewer 1 “Major concerns”:

Finally, a less interesting, less likely, but also possible, explanation for the observed patterns could stem from intra-individual variation in the traits that is not being captured that might influence the assessment of plasticity.....

I think the potential for random error should be acknowledged a little more in the paper. Technical variation is considered and dismissed in several places, but biological variation isn't really discussed.....

We believe we can address the reviewers concern about the effect of intra-individual variation especially how it relates to plasticity. The possibility of intra-individual variation, in particular for WAM, has been added to the discussion on lines 684-724.

Lines 708-715: “Alternatively, unaccounted for within individual variation may explain why there is no significant relationship between 12°C WAM and 28°C WAM in individuals with extreme plasticity. Specifically, measures of WAM at one temperature for individuals with very low or very high plasticity (upper and lower 10% CI) were impacted by some individual biological effect (e.g. stress response). Thus, for these individuals with extreme plasticity there may have been a lack of repeatability in WAM measurement due to biological and not technical variation impacting the ability to assess acclimation response and correlation between performance at 12°C and 28°C.”

Line 270: Should this be Fig S2?

Thank you, yes, this should be a reference to Fig S2.

Line 311: I found this text in combination with Figure 5 to be a bit confusing. The figure shows Cardiosomatic index, and you can only derive the conclusion reported in the text (that the hearts were bigger on an absolute basis) by taking into account the parenthetical statement in the figure legend that body mass was the same. The title of the figure is thus also a little unclear. Consider redoing the figure as two regression lines of heart mass against body mass for the two acclimation temperatures and reporting CSI in the text (i.e. using the presentation from Fig S5).

Figure 5 has been replaced with formerly Figure S3 to clearly show that there was no significant difference in body mass and that there was a significant difference in heart mass between acclimation temperatures. The figure legend has also been edited for clarity and now reads:

“Figure 5: Relationship between heart mass and body mass. Relationship between heart mass (ventricular mass) and body mass for individuals at 12°C (blue) and 28°C (orange) acclimation conditions. $N = 55$ at 12°C, $N = 50$ at 28°C. *Cardiosomatic index at 12°C was significantly greater than at 28°C despite no difference in body mass between acclimation temperatures. P-values from linear regressions, shaded region shows 95% confidence interval.*”

Line 343: I think it would be worth explaining a bit more about the difference between your north reference site and the Dayan et al site. You should make clear the geographic distance

between them. If google maps can be relied on, it looks like the Dayan reference site is about 20km further north than your reference site. This is potentially interesting and worthy of comment, given the steep change in allele frequencies and embryonic phenotypic in this part of the species range. Perhaps there is a very steep cline in CT_{max} in this region, which would be a very cool observation! (and a little different from what was observed by Healy et al (2018) who did not find much difference in CT_{max} across New Jersey, but did find a steep cline in hypoxia tolerance).

Thank you for the suggestion. This has been added to the discussion, now on lines 442-455.

Lines 442-455: “Dayan et al (2019) found evidence of genotypic divergence between the TE and two reference populations (different from the reference populations used here) and a significant difference in CT_{max} after 28°C acclimation in individuals collected from the TE site and the north reference, (Mantoloking, NJ, 40°3'0.02"N, ~25km north of the N.Ref site used here, 39°52'28.0"N (32)). In contrast, Healy et al 2018 found that New Jersey F. heteroclitus populations between latitudes of ~39° and ~40° had little difference in CT_{max} when acclimated to 15°C despite diverging mitochondrial genotypes (60). Yet, the different acclimation temperatures used in these two studies may account for the apparently contrasting evidence for or against a steep cline in CT_{max} along the New Jersey coast for F. heteroclitus. We found that under cooler (12°C) acclimation conditions, inter-individual variation was greater than under warmer (28°C) acclimation conditions, which could explain why a set of populations at similar latitudes did not differ in CT_{max} when measured at 15°C (60) but did differ at 28°C (32). This suggests that that there may be a steep change in CT_{max} along the northern New Jersey coast between 39° and 40° of latitude, which is only measurable under warm (>15°C) acclimation conditions.”

Line 363-365 seems a bit speculative at this point

Yes, this is a bit speculative (now lines 475-476), but gives an alternative view of the data, which we think is important. No change has been made in the manuscript in response to this comment.

Line 560-565: I found this analysis and the discussion of the resulting patterns confusing. I think this needs a bit more clarification, as I am having trouble conceptualizing an underlying mechanism that could account for this pattern.

This section has been re-written and changes including discussion of unaccounted for intra-individual variation should now address this concern (please see above, response to Major Criticisms B).

Line 571: This is one of the places where I think that a discussion of the various alternative ways of looking at the data would be helpful. I think a trade-off is only one possible explanation for the patterns.

Two explanations for these patterns are now provided. 1) variation among individuals in ability to acclimate, and 2) physiological trade-off between acclimation response and low temperature performance (lines 720-724).

Lines 720-724: “As previously discussed, this may be due to variation among individuals in the ability to acclimate, especially when going from 28°C to 12°C acclimation conditions. Alternatively, a trade-off between temperature specific responses and the magnitude of acclimation responses could explain the correlation between low temperature performance and magnitude of acclimation response.”

Line 589: I found the use of the word “obfuscated” interesting. Are you implying that there are actually differences between the populations that you didn’t have power to detect? It seems to me that given your pretty robust sample sizes, if there was actually a biologically relevant difference between these populations, you would have detected it.

Good point, the intention was to highlight that while there were no significant differences among populations, we did find interesting within population variation, not to imply that there are biologically relevant differences among populations that we did not detect in these traits. Word choice has been adjusted to communicate this point more clearly (see below, now lines 741-743).

Line 741-743: “By observing variation in multiple physiological traits among >100 individuals we found that 1) high variation among individuals within populations exceeded variation among populations”...

Reviewer 2:

Firstly, I would like to congratulate you on your dataset, it is rather impressive and it shows that a lot of work went into this project (both in data collection and analysis)! The techniques and data analysis are really interesting and contribute to the know literature. To briefly summarize, your manuscript is looking at the interactions between thermal acclimation and phenotypic plasticity in wild-caught Atlantic killifish. Killifish were sourced from 1) a site known to have higher water temperatures due to anthropogenic heating, 2) a northern reference site, and 3) a southern reference site. Physiological differences did not appear to vary much across population but this was largely attributed to a wide range of inter-individual variation in response to thermal acclimation within the population for a given trait. The inter-individual variation appears to be set by acclimation to colder temperatures (12°C) rather than warmer temperatures. The introduction and discussion suffer from some organizational issues and lack of clarification which I have included in my suggestions below. I only recommend major revision to ensure there is enough time to make the suggested changes. Good luck and I look forward to reading the final manuscript.

* Comments are in order of appearance in the MS, not arranged in order of severity/importance.

** Bolded statements indicate important concerns

Abstract:

Overall the abstract is nicely written and very detailed, however it abruptly ends. Consider adding a final statement about the implications/usefulness of your manuscript (e.g., Studies that focus on population dynamics should account for inter-individual variation that occurs during thermal acclimation because...)

Thank you for the suggestion. We have added to the last line of the abstract
“These findings suggest inter-individual variation in physiological responses to temperature acclimation and, thus, additional research investigating inter-individual is relevant for global climate change responses in many species.”

This addition causes the abstract to exceed the word limit, but we will ask the editor for an exception.

Introduction:

Overall the introduction does an adequate job in presenting the current literature, good job! However the main problem with the introduction is that the information that is presented is not organized or polished. Some simple rearranging of the paragraphs and add details to the ideas that have been introduced will help with this. For more details, look at some of the suggestions below.

Line 34 and various places throughout document. Do not start sentence with because, find some other transition word (e.g., yet, since, however, due too...) This makes the thought following the word because appear more polished/complete.

Thank you. Line 39 and lines 56-58 where “Because” was used at the start of a sentence have been rearranged into more polished statements.

I am not sure how your second paragraph fits into the story of your manuscript. You talk about genetics masking phenotype and how organisms can hide which can result in maladaptive or adaptive responses. But you do not measure any genetic markers or try to ascertain if your populations of animals have maladaptive or adaptive responses. I would suggest removing this paragraph. Then take the third paragraph, divide into sections and expand on the information you have. Try laying your introduction out like this: 1) climate change and anthropogenic heating 2) broad physiological responses 3) acclimation, phenotypic plasticity, and the interactions/drivers 4) specific details about your model organism and study.

This suggestion was considered; however, the second paragraph of the manuscript gives the reader a background in how and why physiological plasticity is important from a broad, population dynamics, perspective and provides a mechanism by which large inter-individual variation in traits important for fitness may be maintained (a major theme of the manuscript). While there are not genomic data presented here, prior evidence suggests that the TE population used in this study has specific genetic differences from other

nearby reference populations providing evidence for local adaptation (discussed on Lines 97-98 and the second section of the discussion lines 433-437). Throughout the discussion we additionally address the evolutionary importance of inter-individual variation and give a review of prior literature on the topic (in particular see lines 414-431, 496-509, 689-703). In addition, there is a rich literature investigating the genetic basis of physiological traits in the species studied here (Reviewed in Crawford et al 2020, but also see references 8, 10, 23, 25, 26, 28, 29, 43, 45, 60, 67 in the manuscript) making this introductory information relevant and interesting to many readers and necessary for readers to interpret the discussion of the evolutionary implications of data presented in the manuscript. For these reasons we would like to maintain the introduction in its current form.

Line 68: Make this sentence into two statements. Here we propose a,b,c. We expect to find x,y,z. These statements could be the lead into your final paragraph.

Respectfully, while several suggestions from the reviewer have improved the manuscript, we have not made changes in response to this comment as it reflects stylistic preference of the reviewer.

Line 76: Is it CTMax or CTmax? I have seen it published both ways, but more frequently and recently as

CTmax. Both abbreviations mean the same thing. Just be aware that more recent studies are using the

CTmax abbreviation, possibly to distinguish between CTmin

While both are appropriate acronyms for critical thermal maximum, we have changed CTMax throughout the manuscript to be CT_{max} to be more consistent with recent literature.

Line 78: None of your listed references (24-26) talk about CaM (what is this abbreviation for? Its unclear) substrates and their relevance as a physiological marker for thermal studies. You need a few lines of justification here to explain what they are and how they relate to cardiac remodeling, which you later talk about in your discussion. In generally animal metabolic rate, CTmax and changes in the concentration of metabolic substrates need to be discussed more since you are using these parameters for the basis of your analysis. This would be good to add to that broad physiological response paragraph (see above). Endogenous... what do you mean by that and how is that relevant?

-Edited to add that this section made sense once I got to the methods, but prior to reading it was really unclear and confusing. Still consider elaborating this section in the introduction. Something along the lines of cardiac tissue metabolic in the presence and absence of aerobic substrates and then reference your methods paper.

CaM is now defined in the abstract and a sentence has been added to the introduction (line 83-89) to provide more context for the reader before they get to the methods, which we are grateful to know have sufficient detail.

Lines 83-89: "Six physiological traits known to be temperature sensitive – whole animal metabolic rate (WAM), critical thermal maximum (CT_{max}) and cardiac metabolic rate (CaM, oxygen consumption of heart ventricles in the presence of glycolytic and non-glycolytic aerobic substrates) were measured in individuals from three populations (28). For CaM, we measured heart tissue oxygen consumption in the presence of four aerobic substrates: glucose, fatty acids, lactate plus ketones plus ethanol, and endogenous [i.e., non-glycolytic metabolism with no added substrates] (29-31)."

Your 3rd paragraph is rather large and would be easier to digest if broken up (see above) and more details/justification was added. Line 89: when you introduce the aims of your project you should also state how you are going to do this. For example we aim to investigate the variation within and among populations by.... And so on with your subsequent aims.

To make the final paragraph more digestible, a break was added at line 98.

Line 98: Assuming reasonably heritability... Hmm isn't this a big assumption here considering that the heritability of these traits was not studied in killifish so you can't be too sure. Also at least one of the listed references suggest that heritability can be low or largely explained by other factors. Essentially there is not a clear cut answer on this unless directly studied in your model organism. I would remove this statement.

Yes, this is an assumption, as stated in the manuscript. However, previously mRNA expression patterns (which are highly heritable, for example Ishikawa et al 2017) have explained up to 81% of variation in *F. heteroclitus* cardiac metabolic rate (Oleksiak et al 2005), CT_{max} in *Fundulus heteroclitus* has been correlated with >50 independent outlier loci in a large investigation of the genetic basis of traits relevant to climate change adaptation (Healy et al 2018), and metabolic rate across many animal taxa including species of teleost fish is heritable (Pilakouta et al 2020, Pettersen et al 2018, Ronning et al 2005, Sadowska et al 2005). There is less evidence on the genetic basis of phenotypic plasticity in animals, yet, Dayan et al 2015 found differential mRNA expression among *F. heteroclitus* from hot and cold adapted populations and further found that these differentially expressed genes differed from those important for acclimation to 12°C and 28°C, suggesting that genes important for thermal acclimation are not the same as those important for local adaptation to variation in habitat temperature. Together, this literature demonstrates that the assumption made here, that metabolic and thermal tolerance traits we measured have a reasonable heritability in this species, is reasonable. We would suggest leaving the statement in as it allows for discussion of the broader implications of our results.

To clarify this to the reader, additional references mentioned here have been added to the manuscript (now line 109).

Ishikawa, Asano, et al. "Different contributions of local- and distant- regulatory changes to transcriptome divergence between stickleback ecotypes." *Evolution* 71.3 (2017): 565-581.

Oleksiak, Marjorie F., Jennifer L. Roach, and Douglas L. Crawford. "Natural variation in cardiac metabolism and gene expression in *Fundulus heteroclitus*." *Nature genetics* 37.1 (2005): 67-72.

Healy, Timothy M., et al. "Tolerance traits related to climate change resilience are independent and polygenic." *Global Change Biology* 24.11 (2018): 5348-5360.

Pilakouta, Natalie, et al. "Multigenerational exposure to elevated temperatures leads to a reduction in standard metabolic rate in the wild." *Functional ecology* 34.6 (2020): 1205-1214.

Pettersen, Amanda K., Dustin J. Marshall, and Craig R. White. "Understanding variation in metabolic rate." *Journal of Experimental Biology* 221.1 (2018).

Rønning, Bernt, Børge Moe, and Claus Bech. "Long-term repeatability makes basal metabolic rate a likely heritable trait in the zebra finch *Taeniopygia guttata*." *Journal of Experimental Biology* 208.24 (2005): 4663-4669.

Sadowska, Edyta T., et al. "Genetic correlations between basal and maximum metabolic rates in a wild rodent: consequences for evolution of endothermy." *Evolution* 59.3 (2005): 672-681.

Dayan, David I., Douglas L. Crawford, and Marjorie F. Oleksiak. "Phenotypic plasticity in gene expression contributes to divergence of locally adapted populations of *Fundulus heteroclitus*." *Molecular ecology* 24.13 (2015): 3345-3359.

Methods

Line 104: Was this all from the same estuary system? You stated in your intro that estuaries can be highly variable from season to season but also among different estuaries.

The reference populations are ~5-10km from the TE population but are all part of a single estuary system and are not known to differ in ecology or abiotic factors besides temperature. Figure 1 shows that all 3 populations are within Barnegat Bay (also see text lines 89-94), which would be defined as a single estuary connecting freshwater inputs from the New Jersey coast to the Atlantic Ocean.

Line 109: Common garden is not a term that is typically found in physiology papers, may want to define this for non-genetic readers.

This has been defined for clarity.

Lines 123-125: "Here, common gardening refers to acclimation to common temperature, salinity, and light cycle to remove the reversible effects of acclimation to local environmental conditions, which may be present in individuals collected from different populations."

Line 133: Consider moving calculations/equations to the statistics section rather than embedded within the experimental methods.

For clarity and logical flow of the methods we have elected to keep the calculations/equations for metabolic rate calculation within the relevant subsection of the methods.

Line 146: Probably should justify the rate of temp change, that it was what was used in previously published studies and has been shown to prevent lag between body and water temperature. Becker, C. D. & Genoway, R. G. Evaluation of the critical thermal maximum for determining thermal tolerance of freshwater fish. *Environmental Biology of Fishes* 4, 245–256 (1979).

Thank you for the suggested reference. This has been added to the CT_{max} methods section.

Line 164-165: “The rate of temperature change is consistent with other published studies and prevents lag between body and water temperature during CT_{max} measurement (43).”

Calculations and statistical analysis seem sound and follow standard methods.

Results

Line 224: You did not talk about how water parameters were collected when you collected fish, so this paragraph comes out of left-field. Consider adding a brief section in your methods that discusses this.

Thank you for the suggestion, this has been addressed on lines 119-120: *“From July-September of 2018 HOBO data loggers were placed at all three sites and used to collect temperature data at a rate of one measure every five minutes.”*

Line 243: Define what you mean by acclimation order in the beginning of the paragraph rather than the end. Reader is left wondering what is acclimation order and how did they miss it and did the author mention it before.

Acclimation order was defined in the methods of the originally submitted manuscript (statistical analysis section) on lines 224-227. No changes were made in reference to this comment.

Lines 224-227: “Within each acclimation temperature there are two groups depending on whether they were acclimated to 12°C first then 28°C or vice versa: group 1 that was acclimated to and assayed at 12°C first before being acclimated to and assayed at 28°C and group 2 that was acclimated and assayed at 28°C first before being acclimated and assayed at 12°C.”

Line 253: variation patterns are acclimation at 12°C and a low response will have a high response at 28°C and vice versa... did these responses correlate with catch site?

Population did not have a significant effect on physiological plasticity, which is now made clear with the addition of the linear models for plasticity in WAM and CT_{max} . These results are in Table S1, discussed on lines 286-291 and 311-315 of the results, and lines 535-583 of the discussion.

Line 301 states CaM was unaffected by acclimation temperature but line 322 states that CaM was one of the more variable traits? How do you reconcile this, these statements seem mutually exclusive? Is it because less variation was explained by thermal acclimation in the CaM method compared to the others? If this is so, this needs to be clarified.

We (the authors) believe that there may be a simple misunderstanding; acclimation effects were not included in measure of inter-individual variation as shown in Table 1. Instead, CaM variation was measured as the standard deviation/mean (coefficient of variation, CV) within each acclimation temperature for each substrate. To clarify this we now state:

Lines 341-345: "As shown by the large confidence interval when comparing 12°C and 28°C groups, cardiac metabolic rate variation within an acclimation temperature among individuals was high (CV=22% to 54% for mass corrected CaM, Table 1)."

The statement that CaM was one of the more variable traits is reflective of the differences among individuals within a given substrate and temperature, and these values can be readily compared to temperature specific CV for other traits in Table 1.

2nd: As stated,

Line 333-334: "Except for endogenous metabolism, CaM was unaffected by acclimation temperature (Fig. 5). Overall, only 0.47% of cardiac metabolic rate variation was explained by temperature."

Except for CaM END, the mean cardiac metabolic rates are similar between acclimation temperatures and acclimation temperature explains less than 0.5% of CaM variance.

We hope this clarifies Reviewer 2 questions.

Discussion

Summary of the first paragraph: despite prior evidence for divergence, there was no evidence in this population and the authors are unsure about the cause. It could be related to migration (although less so), thermal difference is not great enough to illicit change (although less so), traits are difficult to evolve, not selectively important, or trait variation is important for survival. The way to figure out which of these it is, is by doing a genetic analysis.

-Would generation and epigenetics play a role in trait divergence and lack of divergence? Might want to consider these articles and incorporating them into your first paragraph.

- 1) Burggren, W. (2016). Epigenetic inheritance and its role in evolutionary biology: re-evaluation and new perspectives. *Biology*, 5(2), 24. 2)
- 2) Ho, D. H., & Burggren, W. W. (2010). Epigenetics and transgenerational transfer: a physiological perspective. *Journal of Experimental Biology*, 213(1), 3-16.

Thank you for the suggested literature. These ideas have been incorporated on lines 479-488 of the discussion.

Lines 479-488: "This could include epigenetic effects if variation in a trait can be attributed to epigenetic changes within a generation (e.g. DNA methylation) as well as heritable genetic and epigenetic changes across generations (e.g. single nucleotide polymorphisms, maternal effects) (69, 70). To examine epigenetic effects, DNA methylation patterns (for example) in genetically similar individuals who also have divergence in physiological traits could be compared. To address the unexpected lack of divergence among populations, genes associated with the high interindividual variance could be identified and patterns in polymorphisms in these genes partitioned among populations. DNA methylation data in combination with genetic data could also be used to identify single nucleotide polymorphisms important for explaining variation in methylation patterns important for physiological trait variation (i.e. meQLT analysis, (70))."

-This paragraph could be organized to flow a little bit better. After talking about divergence and lack of divergence consider, talking about why that is (the five suggestion layed out above), and then go into details about the likelihood of each possible.

Thank you for the suggestion, the formatting of this section has been altered as suggested to improve flow. Namely, on lines 458-462 the five possible explanations for a lack of trait divergence are listed with the remaining text of the paragraph dedicated to the likelihood of each.

Second paragraph of discussion: Consider add an intro sentence at the beginning of the first paragraph under each header. Remind the reader what is inter-individual variation and why it is significant then talk about what you found. The first half of this section is a retelling of the results section, which the reader can skip and beginning at the second paragraph and not miss much since it is a retelling.

In response to reviewer comments, the discussion as a whole has been reformatted. The first paragraph now states interesting and novel results, followed by a paragraph/section discussing interindividual variation, which begins with a summary of relevant results and then goes into a validation of the methods (i.e. discussion of technical variation). To lead into this second paragraph we have added a short sentence to define interindividual variation and its importance.

Lines 394-395: "Quantifying interindividual variation (variation among individuals) has allowed for variation in acclimation response within and among complex traits to be distinguished (48)."

Line 388: when you say reported by other teleost fish...immediately follow that with all the reference you are about to get into detail with. Reported by other teleost fish (reference number to reference number). You need to support that idea that is has been previously reported before getting into the

details.

Thank you, two references have been added here to point the reader to relevant literature on interindividual variation in metabolic rate (now line 417).

Line 392: why is greater individual variation important for evolutionary history? I rather read about more about that, then validation of the technique which happens in the subsequent sentences. Considering making a separate header section (1st paragraph of discussion) that goes into the validation of this technique and why your results are due to biological variation, get that out of the way first. Then the following paragraphs can be spent telling the reader why this is exciting and what it means from an evolutionary perspective.

Thank you for the suggestion, the discussion has been rearranged to first address technical validation and then moves into the remaining discussion sections.

Line 470:....what is your evidence for cardiac remodeling. This is the first I recall hearing about it. You need to reference back to your data and build a case for it or use previously published studies that support this. Hey I found the evidence that supports this at Line 508! So you should probably introduce that sooner.

This line has been edited to introduce the relevant data earlier for clarification. Now lines 531-534.

Lines 531-534: "Additionally, we find evidence of cardiac remodeling (significant difference in heart mass between acclimation temperatures despite no difference in body mass, Fig. 5) that may allow individuals acclimated to different temperatures to maintain cardiac output, which is important for tissue oxygen delivery and whole animal metabolic processes maintenance."

Paragraph at 505: I think I recall reading that you were not able to collect mass for all hearts. Which is a real shame. That would mean that your CaM is not normalized/standardized to tissue mass then? What is CaM standardized too? I only ask because if that is the case you would definitely know the answer to the statement in line 510.

As described on Lines 317-323, body mass was used as a covariate to remove mass effects on cardiac metabolic rates. Heart mass was not measured for 3 individuals, so the sample size was not substantially impacted by this. Additionally, body mass was significantly correlated with heart mass (Fig. 5) and although hearts were significantly smaller at 28°C, using heart mass residuals instead of body mass residuals did not change the results (lines 344-348, Fig. S3).

Line 533: Hmm while some species have a limited capacity to alter metabolic rate others do not.
1) KIRBY, A. R., CROSSLEY, D. A. & MAGER, E. M. 2020. The metabolism and swimming performance of sheephead minnows (*Cyprinodon variegatus*) following thermal acclimation or acute thermal exposure. *Journal of Comparative Physiology B*.

- 2) PILAKOUTA, N., KILLEN, S. S., KRISTJÁNSSON, B. K., SKÚLASON, S., LINDSTRÖM, J., METCALFE, N. B. & PARSONS, K. J. 2020. Multigenerational exposure to elevated temperatures leads to a reduction in standard metabolic rate in the wild. *Functional Ecology*, 34, 1205-1214.
- 3) SANDBLOM, E., GRÄNS, A., AXELSSON, M. & SETH, H. 2014. Temperature acclimation rate of aerobic scope and feeding metabolism in fishes: implications in a thermally extreme future. *Proceedings of the Royal Society of London B: Biological Sciences*, 281, 20141490.
- 4) MCDONNELL, L. H. & CHAPMAN, L. J. 2016. Effects of thermal increase on aerobic capacity and swim performance in a tropical inland fish. *Comparative Biochemistry and Physiology Part A: Molecular & Integrative Physiology*, 199, 62-70.
- 5) RUMMER, J. L., COUTURIER, C. S., STECYK, J. A. W., GARDINER, N. M., KINCH, J. P., NILSSON, G. E. & MUNDAY, P. L. 2014. Life on the edge: thermal optima for aerobic scope of equatorial reef fishes are close to current day temperatures. *Global Change Biology*, 20, 1055-1066.

Thank you for the suggested references. This sentence (now line 645) has been updated to provide examples of species that do and do not have a limited capacity to alter metabolic rate, in particular in response to temperature acclimation.

Now line 644-646: "Other data also suggest a limited ability to compensate metabolic rate through acclimation in many fish species (29, 84-86) although see (87-89)), including in response to cold acclimation in F. heteroclitus (84)."

Paragraph at 539: This is really novel and should be discussed/highlighted more. Particularly mention this in the abstract instead of buried in the discussion. And your novel findings want to be the bulk of your discussion not validation of experimental methods. This paper about the sheepshead minnow an ecologically similar species may be helpful here. FANGUE, N. A., WUNDERLY, M. A., DABRUZZI, T. F. & BENNETT, W. A. 2014. Asymmetric Thermal Acclimation Responses Allow Sheepshead Minnow *Cyprinodon variegatus* to Cope with Rapidly Changing Temperatures. *Physiological and Biochemical Zoology*, 87, 805-816.

We agree that these results are novel and important and have edited the abstract to highlight these findings (lines 24-26) and expanded the discussion (lines 649-657) in light of the provided reference.

Lines 649-657: "Yet, in sheepshead minnow, an ecologically similar estuarine species, thermal tolerance is gained asymmetrically during acclimation to warm or cold temperatures (gain 50% of thermal tolerance when acclimating from ~11°C to ~18°C, (90)). That is, exposure to a different temperature, regardless of the magnitude of increase or decrease in temperature, results in a gain of the majority of physiologically available thermal tolerance (or loss for cold acclimation) within ~20 days. If a similar mechanism for thermal tolerance is present in F. heteroclitus, our evidence of incomplete active acclimation may represent the majority of thermal tolerance accrual available for this species."

Line 587: Conclusion needs a stronger/global impact and ending about the significance of the data within the realm of the published literature. Why the findings are novel.

Thank you for the suggestion, two sentences have been added to the conclusion section to highlight novel findings and broad significance of the data.

Lines 754-759: “Overall, our findings suggest inter-individual variation in trait specific physiological responses to temperature acclimation. We find that some individuals may have a greater capacity for acclimation response than others and demonstrate a need for additional research investigating inter-individual variation in physiological plasticity of complex traits, which are relevant for global climate change response in many species.”